# A recyclable biomass electrolyte towards green zinc-ion batteries

Hongyu Lu[1,2,8], Jisong Hu[3,8], Xijun Wei[1,8], Kaiqi Zhang[4,8], Xiao Xiao[1], Jingxin Zhao [5] ✉, Qiang Hu[6], Jing Yu[7], Guangmin Zhou [1] ✉ & Bingang Xu [5] ✉

The operation of traditional aqueous-electrolyte zinc-ion batteries is adversely affected by the uncontrollable growth of zinc dendrites and the occurrence of side reactions. These problems can be avoided by the development of functional hydrogel electrolytes as replacements for aqueous electrolytes. However, the mechanism by which most hydrogel electrolytes inhibit the growth of zinc dendrites on a zinc anode has not been investigated in detail, and there is a lack of a large-scale recovery method for mainstream hydrogel electrolytes. In this paper, we describe the development of a recyclable and biodegradable hydrogel electrolyte based on natural biomaterials, namely chitosan and polyaspartic acid. The distinctive adsorptivity and inducibility of chitosan and polyaspartic acid in the hydrogel electrolyte triggers a double coupling network and an associated synergistic inhibition mechanism, thereby effectively inhibiting the side reactions on the zinc anode. In addition, this hydrogel electrolyte played a crucial role in an aqueous acid-based Zinc/MnO$_2$ battery, by maintaining its interior two-electron redox reaction and inhibiting the formation of zinc dendrites. Furthermore, the sustainable biomass-based hydrogel electrolyte is biodegradable, and could be recovered from the Zinc/MnO$_2$ battery for subsequent recycling.

The burgeoning development of aqueous Zn-ion batteries (AZIBs) has led to their exploration for broad applications in renewable energy storage, as the sustainability, high safety, low cost, and environmental friendliness of AZIBs can relieve energy shortages and help to protect the environment[1,2]. Metallic Zn is considered to be an ideal anode for rechargeable AZIBs because of its high theoretical capacity (820 mAh g$^{-1}$), low redox potential (−0.762 V vs the standard hydrogen electrode), and low toxicity, and as Zn is a plentiful resource[3]. However, uncontrolled growth of Zn dendrites on Zn anodes leads to a low coulombic efficiency (CE), and the solvation structure of Zn(H$_2$O)$_6^{2+}$ induces a series of side reactions on Zn anodes and metallic Zn

corrosion in aqueous electrolytes, which cause short circuiting and volume expansion, thereby restricting the long-term operation of AZIBs[4]. Thus, there is an urgent need to exploit available and feasible strategies to protect Zn anodes from the above-described processes.

The design and development of functional hydrogel (i.e., the hydrogel has different functional characteristics) electrolytes as replacements for traditional aqueous electrolytes has proven to be effective for inhibiting the growth of Zn dendrites on Zn anodes and the occurrence of side reactions[5,6]. However, many problems remain to be solved, such as volumetric expansion, parasitic reaction, and dendritic growth, and the mechanism by which most hydrogel electrolytes

[1]Tsinghua Shenzhen International Graduate School, Tsinghua University, Shenzhen 518055, P. R. China. [2]State Key Laboratory of Advanced Welding and Joining, School of Materials Science and Engineering, Harbin Institute of Technology, Harbin 150001, P. R. China. [3]School of Optical and Electronic Information, Huazhong University of Science and Technology, Wuhan 430074, P. R. China. [4]School of Marine Science and Technology, Harbin Institute of Technology (Weihai), Weihai 264209, P. R. China. [5]Nanotechnology Center, School of Fashion and Textiles, The Hong Kong Polytechnic University, Hung Hom, Kowloon Hong Kong 999077, P. R. China. [6]School of Materials and Energy, University of Electronic Science and Technology of China, Chengdu 610054, P.R. China. [7]School of Physics, Harbin Institute of Technology, Harbin 150001, P. R. China. [8]These authors contributed equally: Hongyu Lu, Jisong Hu, Xijun Wei, Kaiqi Zhang. ✉e-mail: jingxzhao@polyu.edu.hk; guangminzhou@sz.tsinghua.edu.cn; tcxubg@polyu.edu.hk

inhibit the growth of Zn dendrites on a Zn anode has not been investigated in detail[5]. Consequently, there is a pressing need for the design of multifunctional hydrogel electrolytes capable of exhibiting all of the properties required in practical applications. In addition, there is a lack of a large-scale recovery method for mainstream hydrogel electrolytes and these are typically prepared from non-biodegradable raw materials. This results in large amounts of hydrogel electrolytes remaining discarded in AZIBs, leading to serious environmental and economic problems. Taken together, the above-mentioned problems demonstrate that a novel hydrogel electrolyte must be developed that displays recyclability, biodegradability, and multifunctionality, thereby enabling the sustainable development of AZIBs with long-term cycling stability. Unlike other hydrogel ingredients, chitosan (CS) and polyaspartic acid (PASP) are biomass materials that are ubiquitous in the shells of molluscs, such as snails, and possess good recyclability and biodegradability. Moreover, the CS and PASP skeletons have abundant carboxylic acid and amine groups, which can form hydrogen-bond crosslinks and effectively enhance the mechanical properties of hydrogels.

In this work, we describe the design and fabrication of a CS- and PASP-containing $ZnSO_4$ hydrogel (denoted "CPZ-H") electrolyte, which exhibits high ionic conductivity and good mechanical properties. Specifically, CPZ-H has high robustness, good recyclability, and biodegradability, and it maintains stable electrochemical performance after mechanical damage and subsequent regeneration. Additionally, calculations show that the abundant anionic functional groups in CS and PASP have high adsorption energy for the Zn (002) crystal plane, which is horizontally aligned. This means that a synergistic inhibition mechanism operates in CPZ-H that facilitates equilibrium ion flux and preferred growth of Zn crystals on the Zn (002) crystal plane to cooperatively regulate Zn deposition. Moreover, anionic groups in CS and PASP with high adsorption energies (such as carboxyl and ether groups) possess desolvation and proton-storage abilities, so they can effectively restrict the decomposition of active water molecules and further restrain adverse reactions on the surface of Zn anodes. Therefore, a symmetric cell consisting of bare Zn anodes and CPZ-H (Zn/CPZ-H/Zn) achieved an ultralong cycle life (2200 h) at a high current density (10 mA cm$^{-2}$), and had a high average CE (99.6%) and persistent stability, even at a high depth of discharge (DOD; 80%). $MnO_2$ experiences a two-electron-transfer redox reaction in an acidic Zn-$MnO_2$ system, which can promisingly double the specific capacity of $MnO_2$ from 308 to 616 mAh g$^{-1}$[2]. However, an acidic electrolyte with H$^+$ will cause a strong hydrogen evolution reaction on the surface of the Zn anode, which greatly affects the cyclic life of the acidic Zn-$MnO_2$ battery. Encouragingly, the CPZ-H electrolyte was effective in aqueous-acid Zn/$MnO_2$ batteries and maintained its interior two-electron redox characteristics due to its good compatibility with a Zn metal anode; it also achieved a high initial capacity (523.6 mA h g$^{-1}$) at a current density of 0.1 A g$^{-1}$, owing to its good proton-redistribution behavior and acid-corrosion resistance. Furthermore, a Zn/$MnO_2$ full cell containing the CPZ-H electrolyte exhibited an impressive capacity retention of 92.5% at a current density of 5 A g$^{-1}$ after 5000 cycles without concomitant introduction of excess Mn$^{2+}$ into the electrolyte. Moreover, the CPZ-H electrolyte could be recycled and reused after battery operation: regenerated CPZ-H (rCPZ-H) electrolyte was used to assemble a Zn/$MnO_2$ full cell, and this exhibited good cycling stability (86.7% after 2400 cycles at a current density of 5 A g$^{-1}$). In addition, the CPZ-H electrolyte can be decomposed under natural conditions by microorganisms.

## Results

### Fabrication and investigation of the CPZ-H electrolyte
The design and fabrication process of the CPZ-H electrolyte is illustrated in Fig. 1a. CS and PASP were obtained from snail shells. In contrast to the conventional acid-based method used to prepare CS hydrogel (C-H), a precooled aqueous alkali–urea solution was used to increase the dissolution of CS for the preparation of the CPZ-H electrolyte. Subsequently, as such solutions are thermosensitive, the CPZ-H electrolyte was gelled by heating and solvent exchange (as detailed in the Experimental section)[7]. As mentioned, the chains of CS were well coupled to PASP within CPZ-H via hydrogen bonds, which greatly enhanced the mechanical properties of the CPZ-H electrolyte. A scanning electron microscopy (SEM) image (Fig. 1b) shows the micromorphology of the freeze-dried CPZ-H electrolyte. Extensively layered and interconnected pores are visible, indicating that it had a high specific surface area (Supplementary Fig. 1) that can accommodate more electrolytes and permit free drift of dissolved ions. Figure 1c presents the Fourier-transform infrared spectra of C-H, CPZ-H, and CPZ-H electrolytes without $ZnSO_4$ (denoted as CP-H). The peak at 3210 cm$^{-1}$ is attributable to the O-H and N-H stretching vibrations of CS, with the broad shape due to these groups being involved in hydrogen bonds. The peaks at 2875 and 1082 cm$^{-1}$ are attributable to C-H and C-O-C stretching vibrations, respectively. Moreover, the characteristic peaks of PASP are observed at ~1512 and 1612 cm$^{-1}$, representing the N-H bending vibration and C=O stretching vibration of an amino group, respectively. Furthermore, the characteristic peak of the carboxyl group of PASP shifted from 1398 to 1390 cm$^{-1}$ when $ZnSO_4$ was added into CP-H[8]. As shown in Fig. 1d, CPZ-H had high water retention (80.5%) after 84 h, which was much higher than those of the CP-H (66.2%) and C-H (45.2%) electrolytes, demonstrating that the hydrophilicity of the CPZ-H electrolyte was greater than those of the latter electrolytes. Similarly, the digital images in the insert of Fig. 1d demonstrate that the CPZ-H electrolyte exhibited almost no water-loss-related volume changes after 84 h.

The mechanical properties of the C-H, CP-H, and CPZ-H electrolytes were examined by determining their typical stress–strain curves (Fig. 1e). It was found that the CPZ-H electrolyte possesses a high tensile strength (53.2 kPa), a large fracture strain (1298%), and that its maximum strain was approximately three times that of the C-H electrolyte. To further verify the flexibility of the CPZ-H electrolyte, a series of extreme tests were performed, which revealed that it had excellent flexibility and stretchability, i.e., it did not crack, even when stretched, twisted, and knotted (Fig. 1f, g). Impressively, the CPZ-H electrolyte can also be drawn into a thin thread that holds a weight of 1 kg (Fig. 1h), which is 10,000 times its own weight. In addition, CPZ-H electrolytes of different colors can be molded into various shapes, revealing their good plasticity (Fig. 1i). The ion diffusion coefficients of the CPZ-H electrolyte at various temperatures are illustrated in Fig. 1j, revealing that its ion diffusion coefficient remained stable (at $4.1 \times 10^{-10}$) at 25 °C but substantially increased with heating to 50 °C (due to a moderate amount of water evaporation) and remained at this level for ~1 h due to thermal effects[6]. However, further continuous evaporation meant that the remaining water molecules within the CPZ-H electrolyte were insufficient to enable the effective transport of ions, such that the ion diffusion coefficient decreased significantly to $2.4 \times 10^{-11}$. However, the ion diffusion coefficient gradually increased when the CPZ-H electrolyte was placed in a room-temperature environment. Therefore, controlling the water content of the CPZ-H electrolyte by varying the temperature can regulate the migration of ions within its hydrogel structure. Supplementary Fig. 2 shows the ionic conductivity of the CPZ-H and CZ-H electrolytes at various temperatures of −20 to 30 °C, and the ionic conductivity of the CPZ-H electrolyte is greater than that of the CZ-H electrolyte due to the presence of PASP in the CPZ-H electrolyte, which allows it to form more porous structures via crosslinking than that of the CZ-H electrolyte. Moreover, the carboxyl groups on the surface of PASP attract counter-ions and thus are served as additional hopping sites for ion transfer[9]. After 15 days of soaking in an aqueous solution of sulfuric acid (pH ~1), testing indicated that the CPZ-H electrolyte maintained its structural integrity, indicating its good resistance to corrosion in highly acidic conditions (Fig. 1k–n).

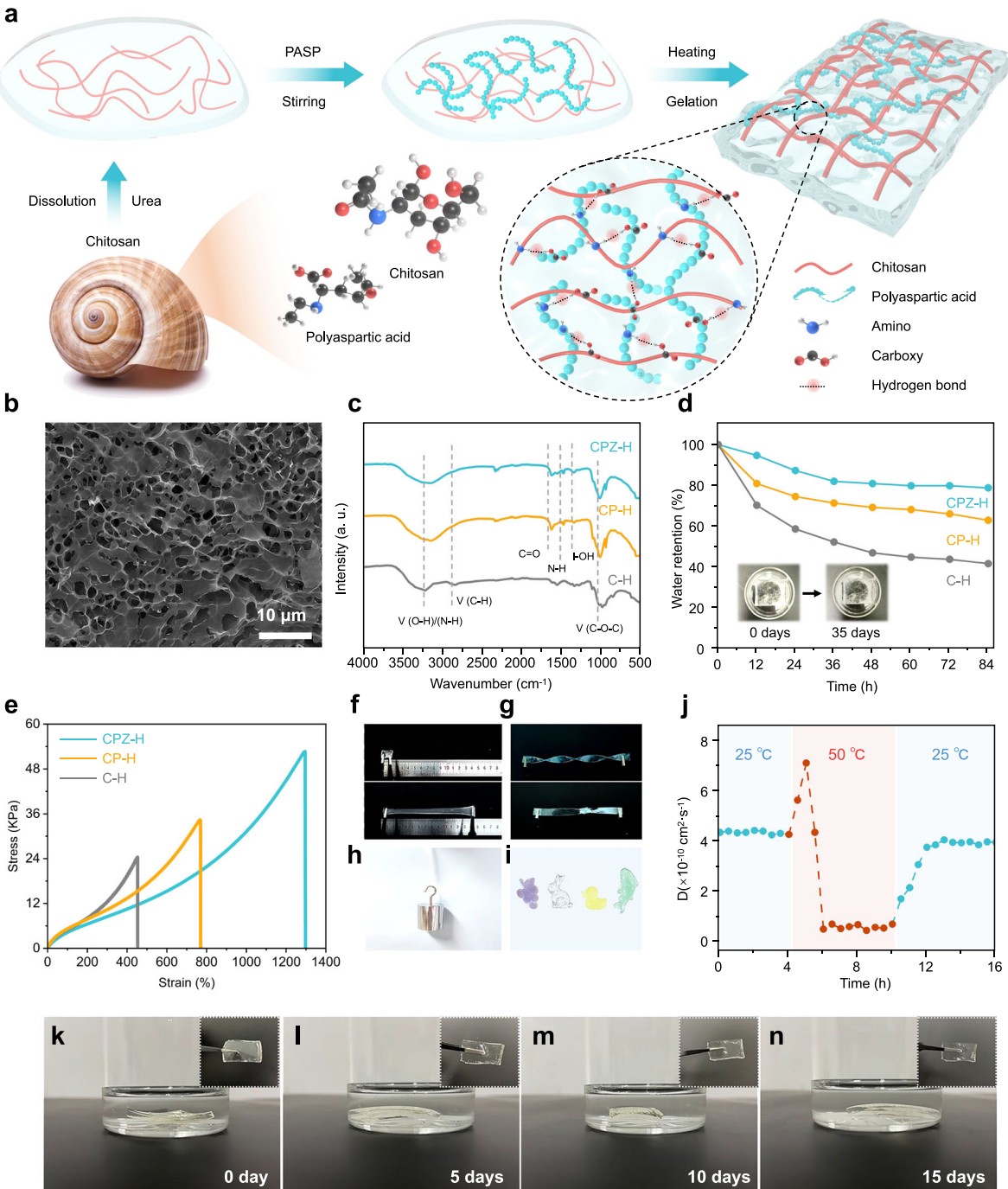

**Fig. 1 | Fabrication and investigation of the CPZ-H electrolyte. a** Schematic diagram of the CPZ-H electrolyte preparation process. **b** SEM image of CPZ-H electrolyte. **c** FTIR spectra, **d** Water-retention tests, and **e** tensile tests of C-H, CP-H, and CPZ-H electrolytes. **f–h** Optical photographs of original, twisted, strained, knotted, and load-holding of CPZ-H electrolytes, and **i** CPZ-H electrolyte shaped into various patterns. **j** Ion diffusion coefficient of the CPZ-H electrolyte as the temperature was changed from 25 to 50 °C. **k–n** Optical photographs of the CPZ-H electrolyte in an acid-soaking experiment that lasted for 15 days.

## Investigations of electrochemical properties and Zn deposition behavior of the Zn/CPZ-H anode

To confirm the feasibility and unique advantages of using the CPZ-H electrolyte to protect a Zn anode, a hydrogel electrolyte consisting of CS and ZnSO$_4$ was also fabricated and was denoted the "CZ-H" electrolyte. Then, the electrochemical properties of various Zn anodes in symmetric cells were systematically investigated. As shown in Fig. 2a, the bare Zn anode had a cycle life of only 190 h with a voltage hysteresis of 436 mV, prior to short circuiting at a current density of 10 mA cm$^{-2}$. However, compared with the bare Zn anode and the Zn/

CZ-H anode, the Zn/CPZ-H anode had a lower voltage hysteresis (96 mV); moreover, its cycle life (2200 h) was more than ten times that of the bare Zn anode. Furthermore, the Zn/CPZ-H anode displayed superior stability and reversibility, even under severe conditions (25 mA cm$^{-2}$ and 25 mAh cm$^{-2}$), revealing that the addition of PASP significantly assisted in suppressing the growth of Zn dendrites and the hydrogen evolution reaction (HER) that competes with Zn deposition (Fig. 2b). The rate performance of the symmetric cells containing Zn anodes (Fig. 2c) illustrated that the Zn/CPZ-H anode had a stable voltage profile, with lower voltage hysteresis when the current density

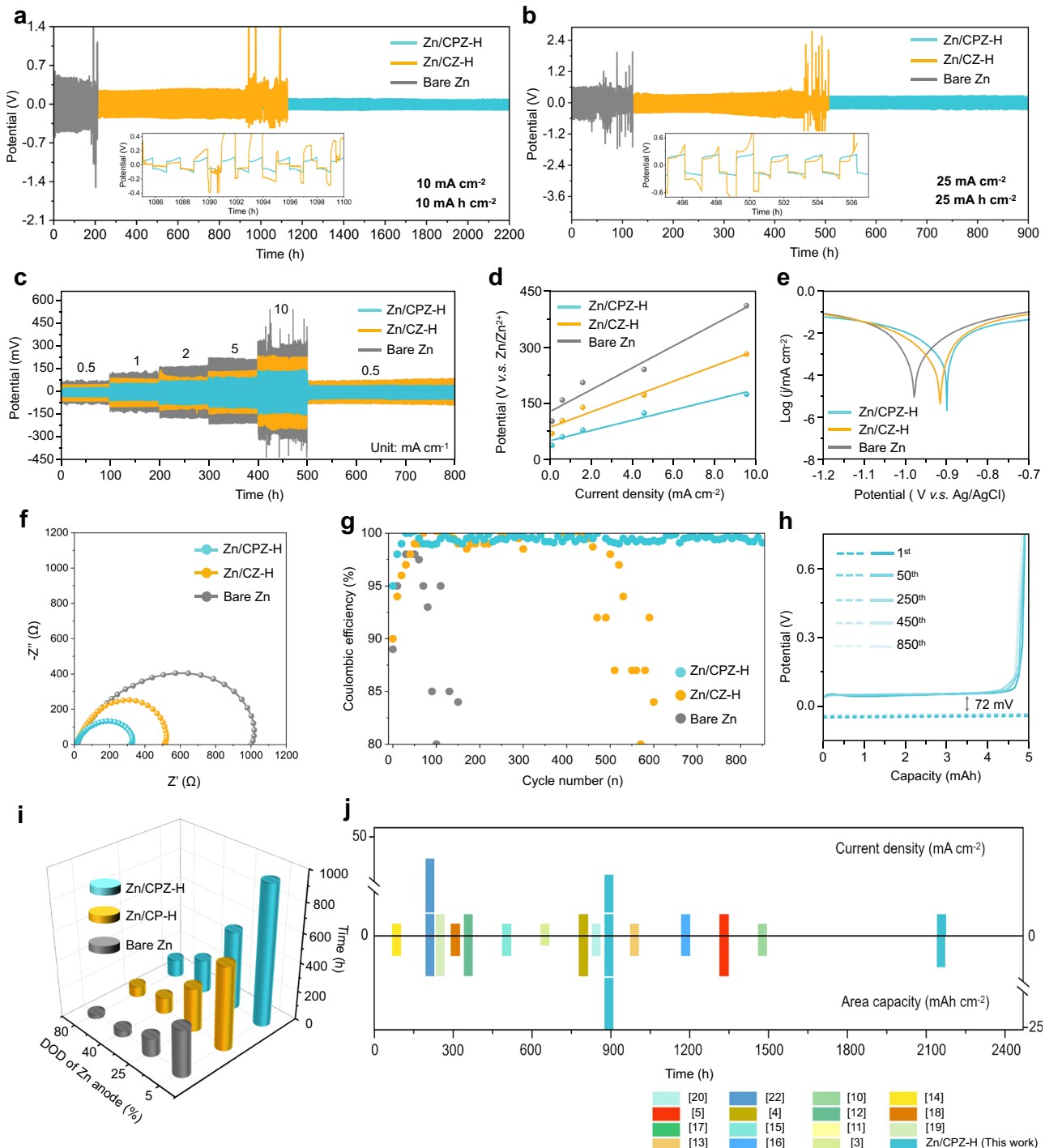

**Fig. 2 | Investigations of electrochemical properties of the Zn/CPZ-H anode.** **a** Voltage profiles of symmetric cells with Zn/CPZ-H, Zn/CZ-H, and bare Zn anodes at current densities of 10 and **b** 25 mA cm⁻². **c** Rate performance of symmetric cells with Zn/CPZ-H, Zn/CZ-H, and bare Zn anodes at current densities from 0.5 to 10 mA cm⁻². **d** Exchange current density curves at various rates in symmetric cells.

**e** Tafel curves, **f** Nyquist plots, and **g** Coulombic efficiencies of Zn/CPZ-H, Zn/CZ-H, and bare Zn anodes. **h** Voltage profiles of a Zn/CPZ-H anode at 10 mA cm⁻². **i** DOD of bare Zn and Zn/CPZ-symmetric cells. **j** Comparison of cyclic reversibility with cells that have been recently reported[3–5,10–22].

was increased from 0.5 to 10 mA cm⁻², while the bare Zn anode abruptly failed at 10 mA cm⁻². This is further evidence that the synergistic interaction between CS and PASP is critical for the increased cycling stability of the Zn/CPZ-H anode. Figure 2d shows that the Zn/CPZ-H anode had an exchange current density of 0.378 mA cm⁻², which is lower than those of the bare Zn and Zn/CZ-H anodes, demonstrating its good corrosion resistance. As summarized in Fig. 2e, the corrosion rates of the bare Zn, Zn/CZ-H, and Zn/CPZ-H anodes were surveyed by

linear polarization measurements. The Zn/CPZ-H anode retained a corrosion current of 0.27 mA cm⁻², which is less than those of the Zn/CZ-H and bare Zn anodes, reflecting the distinct anti-corrosion properties of the CPZ-H electrolyte.

As revealed by the Nyquist plots in Fig. 2f, the impedance of the bare Zn anode attests to its high interfacial charge-transfer resistance (1027 Ω). However, the Zn/CPZ-H and Zn/CZ-H anodes had lower interfacial charge-transfer resistances (316 and 511 Ω, respectively),

indicating that the functional CPZ-H electrolyte achieved the fastest charge transfer. Supplementary Fig. 3 reveals the cyclic voltammetry (CV) curves of the bare Zn and Zn/CPZ-H anodes; as can be seen, the Zn nucleation overpotential of the Zn/CPZ-H anode was less than that of the bare Zn anode, which demonstrates that there were faster nucleation kinetics and a lower nucleation barrier during the Zn deposition process in the Zn/CPZ-H anode than during that in the bare Zn anode. Encouragingly, the Zn/CPZ-H//Cu cell operated over 850 cycles had an average CE of 99.6% at a current density of 10 mA cm$^{-2}$ with a limited capacity 5 mAh cm$^{-2}$ (Fig. 2g), indicating the excellent reversibility and durability of the Zn/CPZ-H anode. Additionally, the charge–discharge voltage profiles from the 1st to 850th cycles are presented in Fig. 2h and Supplementary Fig. 4. The Zn/CPZ-H/Cu cell had a plating/stripping voltage hysteresis of 72 mV, which was substantially lower than those of the Zn/CZ-H anode (109 mV) and bare Zn anode (153 mV). The DOD of the Zn anodes was also evaluated, as it is an important metric for revealing the suitability of AZIBs for use in practical applications. As observed in Fig. 2i and Supplementary Fig. 5, the bare Zn anode short-circuited after ~300 h when its DOD was 5%, leading to a low Zn utilization rate and energy density. In contrast, the lifespan of the Zn/CPZ-H anode was prolonged to 120 h, even under the harsh condition of 80% DOD, owing to the protection afforded by the CPZ-H electrolyte. In addition, a Zn/CPZ-H//Zn/CPZ-H symmetric cell displayed satisfactory cyclic performance and reversibility in an alternating test of continuous running and shelving (Supplementary Fig. 6), indicating that the Zn/CPZ-H anode possesses tremendous potential for recovery performance. Furthermore, this ultrahigh cycling life of the Zn/CPZ-H anode exceeds the cycling lives of most of the modified Zn anodes that have been previously reported (Fig. 2j)[3–5,10–22].

To further investigate the ability of the CPZ-H electrolyte to modulate the Zn deposition behavior of the Zn/CPZ-H anode, a series of ex situ characterizations were performed. The ex situ SEM images of the bare Zn anode in Fig. 3a–c reveal that the surface of the bare Zn anode was covered with Zn dendrites after 100 and 200 h. These formed disorderly piles of Zn flakes, which accumulated on the bare Zn anode and eventually led to battery failure. Supplementary Fig. 7 shows SEM images of the Zn/CZ-H anode at various cycle times, clearly demonstrating that the excessive growth of Zn dendrites was somewhat suppressed, due to the CS in the CZ-H electrolyte effectively regulating Zn$^{2+}$ flux. However, the CS only delayed the growth rate of dendrites, and it could not completely eliminate the damage to the separator caused by dendrite growth. Thereby, abundant dendrites were present on the surface of the Zn/CZ-H anode at 1100 h. In contrast, a large number of Zn dendrites grew on the Zn (002) plane on the surface of the Zn/CPZ-H anode over a cycling time of 2200 h (Fig. 3d–f), which is attributable to the stronger adsorption energy of PASP along the Zn (002) plane, indicating that the PASP in the CPZ-H electrolyte enhanced deposition of Zn onto the Zn (002) crystal plane during the plating/stripping process (Supplementary Fig. 8). The cross-sectional SEM images of the bare Zn and Zn/CPZ-H anodes are shown in Supplementary Figs. 9, 10, respectively; these reveal that vertical Zn dendrites were present on the bare Zn anode but not on the Zn/CPZ-H anode, due to the presence of PASP in the CPZ-H electrolyte. Furthermore, ex situ laser confocal scanning microscopy was performed to survey the surface conditions of the Zn anodes after various cycling times. Figure 3g–i shows that the surfaces of the bare Zn anode initially had a smooth planar morphology, but had accumulated flake-like dendrites after 100 and 200 h. In contrast, Zn nanosheets were uniformly deposited across the entire surface of the Zn/CPZ-H anode, and deposited Zn grew in the horizontal direction with no abnormal local protrusion, even after 2200 h (Fig. 3j–l).

To further scrutinize the mechanism of growth of the deposited Zn layer, chronoamperometry was performed at a constant over-potential of −150 mV. As shown in Fig. 3m, the current density continuously increased after 300 s, which demonstrates that the two-dimensional diffusion process triggered the uneven growth of a Zn film and that Zn$^{2+}$ tended to aggregate and ultimately form dendrites. During this process, deposited Zn atoms would have diffused transversely along the surface to find the most favorable sites for charge transfer, i.e., those with decreased surface energy and exposed areas. In the Zn/CPZ-H anode, the rate of increase in the current density slowed after the initial 50 s, and then a stable and constant three-dimensional diffusion process became dominant. This process increased the number of sites for Zn nucleation and thus led to an increase in the uniformity and density of the deposited layer of Zn (inset of Fig. 3m). In addition to guiding uniform nucleation and Zn deposition, the CPZ-H electrolyte also accelerated the kinetics of Zn$^{2+}$ plating/stripping. Specifically, the activation energy ($E_a$) of Zn$^{2+}$ transference was calculated using the Arrhenius equation, showing that $E_a$ decreased from 29.14 kJ mol$^{-1}$ in the absence of the CPZ-H electrolyte to 18.43 kJ mol$^{-1}$ in the presence of the CPZ-H electrolyte (Fig. 3n and Supplementary Fig. 11). Online electrochemical mass spectroscopy was also used for ex situ monitoring of the H$_2$ flux during the Zn plating/stripping process, to enable accurate quantification of H$_2$ evolution (Fig. 3o and Supplementary Fig. 12). The H$_2$ evolution rate of the bare Zn//Zn anode symmetric cell reached 0.47 μmol/s over a 5-h cycle, whereas the average H$_2$ evolution rate of the Zn/CZ-H//Zn/CZ-H anode symmetric cell was only 0.16 μmol/s, due to the CZ-H electrolyte suppressing Zn$^{2+}$ flux. Surprisingly, the H$_2$ evolution rate in the Zn/CPZ-H anode symmetric cell was negligible (only 0.04 μmol/s over a 5-h cycle), revealing that PASP could further suppress the HER. The theoretical adsorption energies of the carboxyl and ether anionic groups of CS and PASP in the CPZ-H electrolyte for Zn$^{2+}$ are higher than that of H$_2$O for Zn$^{2+}$, revealing that Zn$^{2+}$ is preferentially captured by such anionic functional groups. As the main barrier to charge transfer usually originates from the Zn$^{2+}$ desolvation process in an electrolyte, the desolvation abilities of CS and PASP were further assessed by calculating the desolvation energies (Fig. 3p). The structure of the solvation environment in the CPZ-H electrolyte was simplified as CS-[Zn(H$_2$O)$_5$]$^{2+}$ and PASP-[Zn(H$_2$O)$_5$]$^{2+}$ to explore the dissolution capacity of its different components (Supplementary Fig. 13). The calculations revealed that the desolvation energies of CS-[Zn(H$_2$O)$_5$]$^{2+}$ and PASP-[Zn(H$_2$O)$_5$]$^{2+}$ (−20.46 and −21.52 eV, respectively) are more negative than that of [Zn(H$_2$O)$_6$]$^{2+}$ (−18.59 eV). This highlights the enhanced desolvation ability of the CPZ-H electrolyte and is consistent with the calculated $E_a$ of the symmetric cells. In addition, the lowest unoccupied molecular orbital (LUMO) and highest occupied molecular orbital (HOMO) energy levels of a Zn anode/electrolyte interface have an enormous effect on the thermodynamic stability of a full battery. Therefore, the LUMO and HOMO of H$_2$O, PASP and CS molecules were determined by density functional theory (DFT) calculations. As shown in Fig. 3q, the LUMO energies of PASP (−2.06 eV) and CS molecules (0.76 eV) in the CPZ-H electrolyte are much lower than that of H$_2$O molecules (1.44 eV), revealing that the low LUMO of the CPZ-H electrolyte is capable of suppressing the spontaneous reduction of H$_2$O molecules, i.e., the HER. To explore the mechanism of this suppression, the Gibbs free energies of H$^+$ adsorption ($\Delta G_H$) for the Zn/PASP, Zn/CS, and bare Zn anodes were calculated. This showed that compared with water dissociation on the bare Zn anode, that on the Zn/CPZ-H anode is more difficult, due to the presence of CS and PASP, which leads to a reduction in the proton supply, thereby inhibiting the generation and evolution of H$_2$ (Fig. 3r). As aforementioned, the deposition of Zn onto the bare Zn anode and the Zn/CPZ-H anode can be illustrated as shown in Fig. 3s. The zincophilic functional groups (hydroxyl and N-acetylamino groups) of CS decrease the energy barrier of desolvation and thus facilitate the uniform deposition of Zn in a vertical electric field. Moreover, the PASP in the CPZ-H electrolyte possesses satisfactorily reversible proton-storage ability, which reduces the trend of HER and further inhibits related parasitic reactions. In addition, PASP strongly adsorbs onto the Zn (002) plane, so it can induce Zn dendrites to grow preferentially on this plane

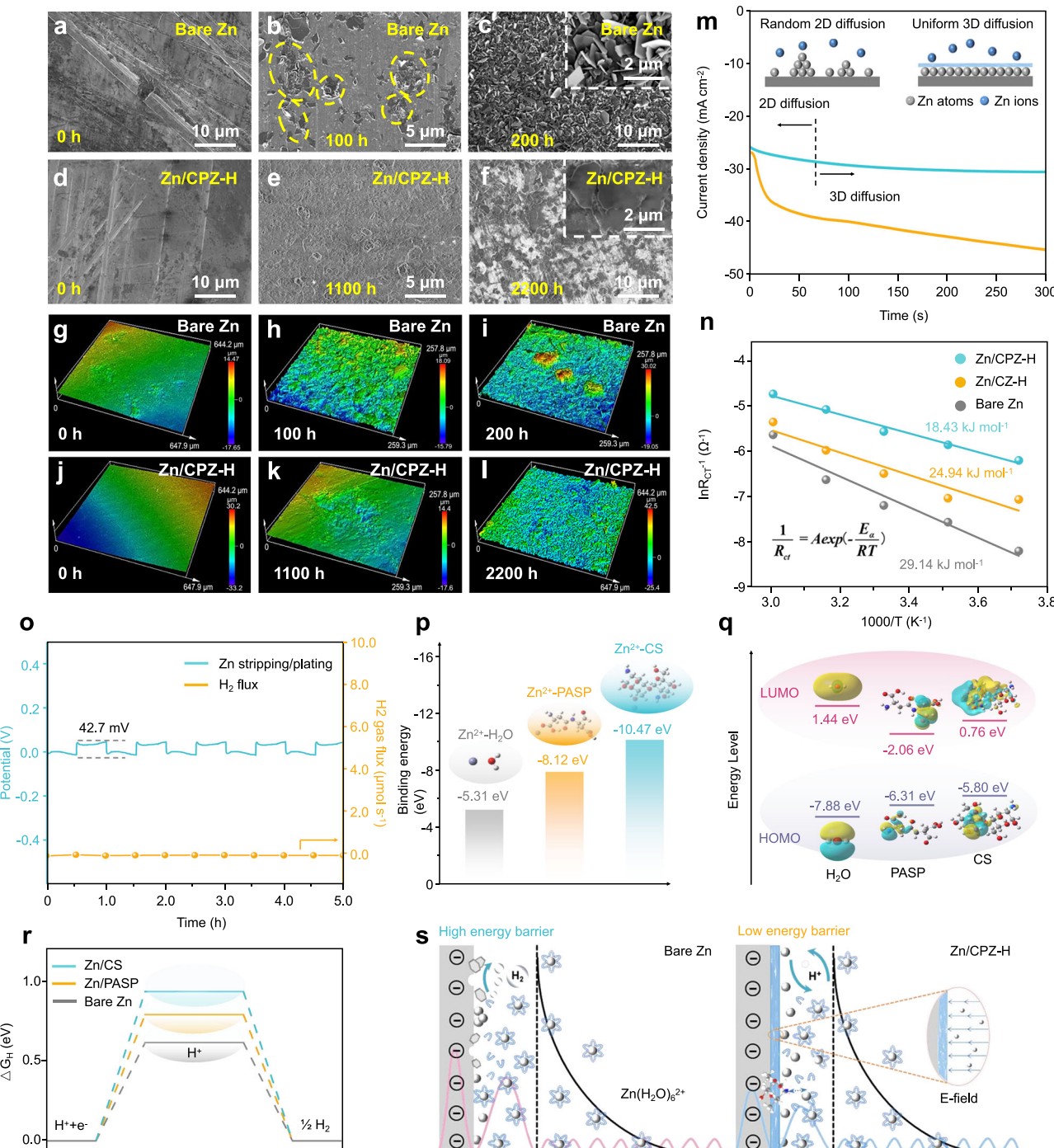

**Fig. 3 | Analysis of Zn deposition behavior of the Zn/CPZ-H anode. a–c** Ex situ SEM images of bare Zn and **d–f** Zn/CPZ-H anodes. **g–i** Ex situ LCSM images of bare Zn and **j–l** Zn/CPZ-H anodes. **m** Chronoamperograms of bare Zn//Zn and Zn/CPZ-H//Zn/CPZ-H symmetric cells at a −150 mV overpotential. Insets: types of $Zn^{2+}$ diffusion and reduction processes. **n** Corresponding Arrhenius curves and comparison of activation energies. **o** Ex situ monitoring of $H_2$ evolution flux of a Zn/CPZ-H anode. **p** Calculated binding energies of $Zn^{2+}$-$H_2O$, $Zn^{2+}$-PASP, and $Zn^{2+}$-CS. **q** Molecular orbital energies of $H_2O$, PASP, and CS molecules calculated by DFT. **r** DFT calculations used to study the $\Delta G_H$ of bare Zn, Zn/CS, and Zn/PASP. **s** Schematic $Zn^{2+}$ deposition processes on bare Zn and Zn/CPZ-H anodes.

(Supplementary Figure 14). The synergistic effect of CS and PASP effectively inhibits the irreversible damage that Zn dendrites can cause to a battery, further increasing the long-term cyclic stability of a battery (Supplementary Fig. 15).

**Electrochemical performance and reaction mechanism of a Zn/CPZ-H//MnO₂ full battery**
Due to the excellent anti-acid-corrosion properties of the CPZ-H electrolyte, CPZ-H immersed in a sulfuric acid solution was used as a

hydrogel electrolyte in an aqueous-acid Zn/MnO₂ battery to stimulate the internal two-electron redox process of the battery. MnO₂ cathode materials were also synthesized (Supplementary Fig. 16), and the corresponding details are described in the Supplementary Information. As shown in Fig. 4a, the ideal chemical reaction in an aqueous-acid Zn/MnO₂ battery with a two-electron redox process delivers a high theoretical capacity (616 mA h g⁻¹) and achieves the maximum energy density of a Zn/MnO₂ battery. The reaction processes that occur on the cathode and anode are shown in Eqs. (1) and (2), and the overall

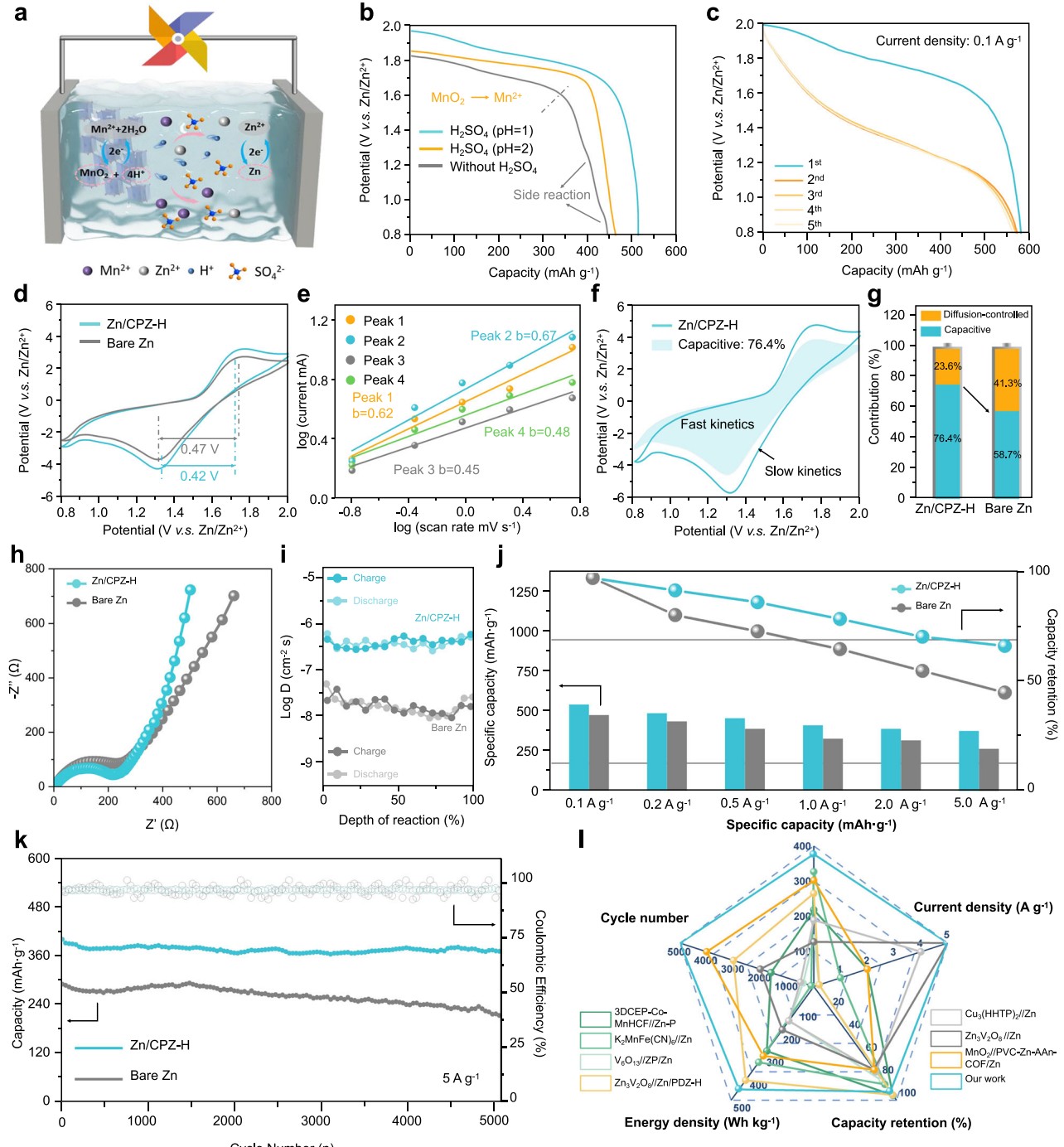

**Fig. 4 | Electrochemical performance and reaction mechanism of a Zn/CPZ-H// MnO₂ full battery. a** Schematic illustration of reactions that occur in an aqueous-acid Zn/CPZ-H//MnO₂ battery. **b** First discharge curves of Zn/MnO₂ batteries containing the CPZ-H electrolyte and various concentrations of H⁺. **c** Discharging capacity of a cell in the first five cycles at a current density of 0.1 A g⁻¹. **d** CV curves of Zn/CPZ-H//MnO₂ and bare Zn//MnO₂ full cells at a scan rate of 2 mV s⁻¹. **e** Plots of corresponding log($i$) versus log($v$) curves of various peaks. **f** CV curve of Zn/CPZ-H// MnO₂ full cell at 5 mV s⁻¹ showing the contribution to the capacitance from fast

kinetic processes and the slow kinetic processes. **g** Comparison of capacitance contributions in terms of **h** EIS curves and **i** Zn²⁺ diffusion coefficients ($D_{Zn^{2+}}$) of Zn/ CPZ-H//MnO₂ and bare Zn//MnO₂ full cells. **j** Rate performance and capacity retention and **k** long-term cycling performance at a current density of 5 A g⁻¹ of Zn/ CPZ-H//MnO₂ full cell. **l** Comparison of the performance of the battery in the current study with the performances that have been reported for other Zn-ion batteries[2,3,5,17,23–33].

reaction process is presented in Eq. (3):

$$Cathode: MnO_2 + 4H^+ + 2e^- \leftrightarrow Mn^{2+} + 2H_2O \quad E^0 = 1.23\,V \quad (1)$$

$$Anode: Zn^{2+} + 2e^- \leftrightarrow Zn \quad E^0 = -0.76\,V \quad (2)$$

$$Overall: Zn + MnO_2 + 4H^+ \leftrightarrow Zn^{2+} + Mn^{2+} + 2H_2O \quad \triangle = 1.99\,V \quad (3)$$

Figure 4b shows that the CPZ-H electrolyte had imperfect discharge performance and side reactions involving Mn²⁺ deposition. Conversely, the pre-addition of protons (H⁺) into the CPZ-H electrolyte

enabled complete charge transfer from $Mn^{4+}$ to $Mn^{2+}$ within the $MnO_2$ cathode. In addition, the discharge capacity of $Zn/MnO_2$ was increased from 480.7 to 533.6 mAh g$^{-1}$ by increasing the concentration of $H^+$ such that the pH decreased from 2 to 1. The first five discharge curves of the $Zn/MnO_2$ cell system at a current density of 0.1 A g$^{-1}$ are shown in Fig. 4c. The $Zn/CPZ$-H$//MnO_2$ full cell delivered a high capacity (523.6 mAh g$^{-1}$) in the first cycle, and the discharge capacity of the second cycle was similar (516.4 mAh g$^{-1}$), which is higher than that of $Mn^{4+}$ to $Mn^{3+}$ in the conventional natural aqueous electrolyte (theoretical capacity = 308 mAh g$^{-1}$). During the subsequent cycle, the $Zn/$ $CPZ$-H$//MnO_2$ full cell achieved high and stable performance during the discharge process, with the single plateau on the discharge curve, indicating that the two-electron reduction from $Mn^{4+}$ to $Mn^{2+}$ was directly realized, i.e., occurred in one step. The CV curves of the $Zn/$ $CPZ$-H$//MnO_2$ full cell in Fig. 4d have a typical single-peak shape with smaller voltage polarization between the oxidation and reduction peaks, and the CV curves of this cell over the first ten cycles (Supplementary Fig. 17) exhibit a similar profile and a pair of fixed redox peaks, revealing the good reversibility of the $Zn/CPZ$-H$//MnO_2$ full cell. A bare $Zn//MnO_2$ full cell was also assembled, using an aqueous electrolyte containing $ZnSO_4$ and $H_2SO_4$, and used as a comparison cell to determine the practical utility of the CPZ-H electrolyte in an aqueous-acid $Zn/MnO_2$ system, The CV curves of the $Zn/CPZ$-H$//$ $MnO_2$ and $Zn//MnO_2$ full cells at scan rates ranging from 0.2 to 5 mV s$^{-1}$ exhibited a pair of well-defined and sharp redox peaks (Supplementary Fig. 18), which again indicates that only one redox reaction occurred in these aqueous-acid $Zn/MnO_2$ batteries. Figure 4e displays the capacitance behavior of the $Zn/CPZ$-H$//MnO_2$ and bare $Zn//MnO_2$ full cells. Significantly, the *b* values of peaks 1 and 2 were 0.62 and 0.67, demonstrating that the reactions represented by these peaks were controlled by both ionic diffusion and capacitive behavior (with the latter being dominant), which would accelerate the reaction kinetics of electrochemical processes in the $Zn/CPZ$-H$//$ $MnO_2$ full cell. Figure 4f, g shows the CV curves of the $Zn/CPZ$-H$//$ $MnO_2$ full cell, indicating that the capacitive contribution ratio at 5 mV s$^{-1}$ (76.4%, as can be seen in the shaded region) was higher than that of the bare $Zn//MnO_2$ cell (58.7%). In addition, the above-mentioned capacitive contribution ratio of the $Zn/CPZ$-H$//MnO_2$ full cell at 5 mV s$^{-1}$ was greater than that at 0.2 mV s$^{-1}$ (40.9%) (Supplementary Fig. 19). This high capacitive contribution is attributable to the fast migration of ions/electrons in the $Zn/CPZ$-H anode, which is also the key requirement for achieving excellent rate capabilities and cycling performance at large current densities.

The increase in kinetics was also demonstrated by electrochemical impedance spectroscopy (EIS). As illustrated in Fig. 4h, the EIS spectrum of the $Zn/CPZ$-H$//MnO_2$ full cell had a greater slope than that of the bare $Zn//MnO_2$ full cell in the low-frequency region, which again illustrates that the former cell featured faster diffusion and transport of $Zn^{2+}$ during the charge/discharge process than the latter cell. Moreover, galvanostatic intermittent titration technique tests showed that the $Zn^{2+}$ diffusion coefficient of the $Zn/CPZ$-H$//MnO_2$ full cell was higher than that of the bare $Zn//MnO_2$ full cell (Fig. 4i and Supplementary Fig. 21). Figure 4j and Supplementary Fig. 20 demonstrate that the $Zn/CPZ$-H$//MnO_2$ full cell had a high capacity (523.6 mAh g$^{-1}$) at a current density of 0.1 A g$^{-1}$ and even a capacity of 376.8 mAh g$^{-1}$ at a current density of 5 A g$^{-1}$, with capacity retention as high as 71.9%. In addition, the capacity of the $Zn/CPZ$-H$//MnO_2$ full cell exhibited almost no decay over 5000 cycles (Fig. 4k), corresponding to an impressive capacity retention rate of 92.5%; and an energy density of 452.16 Wh kg$^{-1}$, even at high power density (753.6 W kg$^{-1}$). Overall, compared with the corresponding performances of most of the previously reported AZIBs, the $Zn/CPZ$-H$//MnO_2$ full cell exhibited a large capacity (523.6 mAh g$^{-1}$), a high energy density (452.16 Wh kg$^{-1}$) and excellent cyclic stability (a capacity retention rate of 92.5% after 5000 cycles), owing to the ionic redistribution and induced planar

deposition of its functional CPZ-H electrolyte (Fig. 4I and Supplementary Information Table 1)[2,3,5,17,23–33].

## Investigation of the $Zn^{2+}$ storage mechanism of the $Zn/CPZ$-H$//$ $MnO_2$ full cell

To further investigate the reaction mechanism of the $Zn/CPZ$-H$//MnO_2$ full cell, a series of ex situ and in situ characterizations were conducted. Ex situ X-ray diffraction revealed the $Zn^{2+}$-storage mechanism of the $Zn/CPZ$-H$//MnO_2$ full cell operating via a two-electron redox process during discharging/charging (Fig. 5a). The $MnO_2$ cathode initially had five main characteristic peaks, all of which shifted to smaller diffraction angles during the first discharge to 0.8 V. Additionally, two extra characteristic peaks were located at 33° and 34°, and correspond, respectively, to the (200) and (103) crystal planes of $Zn_xMnO_2$ and $ZnMnO_4$, respectively. However, there were no peaks corresponding to MnOOH, revealing that $MnO_2$ tended to undergo a two-electron reduction from $Mn^{4+}$ to $Mn^{2+}$. Subsequently, as the battery was fully charged to 2.0 V, the characteristic peaks of $Zn_xMnO_2$ and $ZnMnO_4$ gradually disappeared, and the main peaks of the $MnO_2$ cathode returned to their initial state, which represents the de-intercalation of $Zn^{2+}$ from the $MnO_2$ cathode. Moreover, the galvanostatic charge–discharge profile contained only one plateau, which is further evidence that $Mn^{4+}$ was directly reduced to $Mn^{2+}$ during the electrochemical reaction[34]. Finally, after the second discharge process, the characteristic peaks of $Zn_xMnO_2$ and $ZnMnO_4$ reappeared, and the characteristic peaks of $MnO_2$ were shifted to a smaller diffraction angle, demonstrating the reversibility of the structural change in the $MnO_2$ cathode during the repeated charging/discharging process.

Ex situ X-ray photoelectron spectroscopy (XPS) was performed to further confirm the existence of the two-electron reduction process. Figure 5b presents the spectrum of Mn 2*p* during the charge/discharge process, and the change in the Mn valence is also illustrated by the spectra of Mn 2*p* at various voltages. After the full cell was first discharged to 0.8 V, the intensity of the $Mn^{4+}$ peaks at 654.7 and 642.8 eV gradually weakened, while that of the $Mn^{2+}$ peaks at 653.4 and 642.5 eV was enhanced, which confirms that there was a significant change in the valence of Mn, i.e., from $Mn^{4+}$ to $Mn^{2+}$. Subsequently, after the full cell was charged to 2.0 V, the $Mn^{4+}$ peaks regained their initial intensity and the $Mn^{2+}$ peak gradually weakened. Finally, as the full cell was again discharged to 0.8 V, the intensity of the $Mn^{2+}$ peak was restored, confirming that the two-electron reduction occurred during the discharge/charge process[35,36]. Moreover, the full cell exhibited excellent reversibility. The Zn 2*p*1/2 and Zn 2*p*3/2 peaks gradually increased in size, corresponding to the intercalation of $Zn^{2+}$ into the $MnO_2$ cathode, and with the extraction of $Zn^{2+}$, these peaks decreased in size (Fig. 5c). In addition, the O 1*s* and N 1*s* XPS spectra are illustrated in Fig. 5d, e; the O 1*s* spectrum contains three peaks, corresponding to Mn-O, O-H, and C-Mn-O bonds, respectively. The enhancement of the Mn-O and Mn-N peaks is attributable to $Zn^{2+}$ extraction, increasing the binding energy of the restored $Mn^{4+}$ with electronegative functional groups. Figure 5f and g show in situ Raman spectra of the $Zn/CPZ$-H$//MnO_2$ full cell. The peak at 653 cm$^{-1}$ (*v*) is assignable to the Mn-O stretching vibration in the $d_z^2$ orbital of [$MnO_6$] octahedra. Due to the effect of redox reactions on the Mn-O stretching vibrations, the Mn-O energy band changed significantly when the applied potential was decreased from 2.0 to 0.8 V and then back to 2.0 V. Specifically, the intensity of the Mn-O peak gradually increased when the cell was charged to a higher potential, revealing the presence of an oxidized form of elemental Mn (i.e., $Mn^{4+}$), that is, Mn was oxidized and the Mn-O bonds in [$MnO_6$] octahedra were shortened. Conversely, when the cell was discharged to a lower potential, $Mn^{4+}$ was reduced and the Mn-O bonds of [$MnO_6$] octahedra were elongated, causing the weakening of the Mn-O stretching vibration in the $d_z^2$ orbital of [$MnO_6$] octahedra[37]. This further demonstrates the reversibility of the $Zn/CPZ$-H$//MnO_2$ full cell during the charging/discharging process. The $Zn^{2+}$-intercalation/

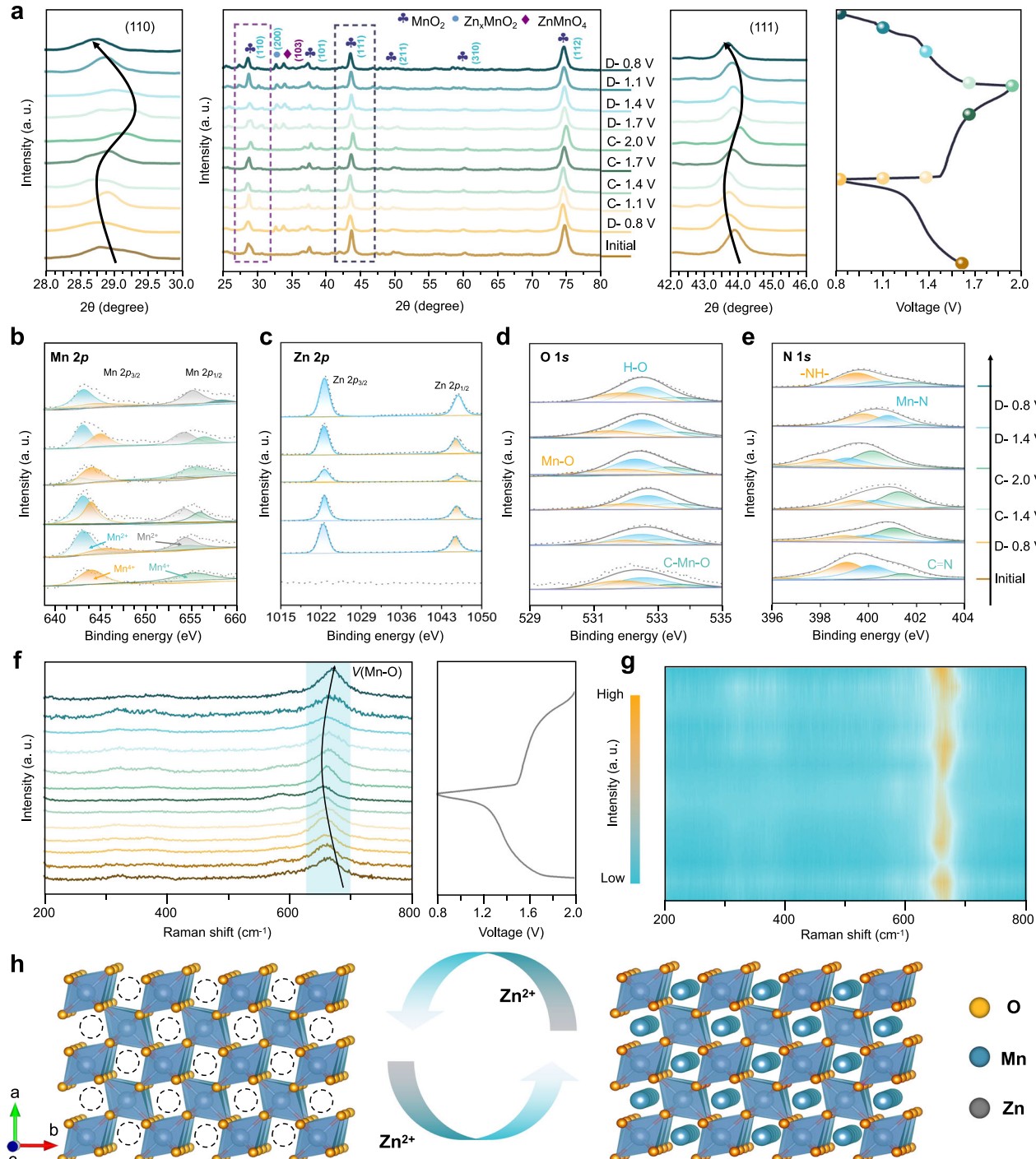

**Fig. 5 | Investigation of the Zn²⁺ storage mechanism of the Zn/CPZ-H//MnO₂ full cell. a** Ex situ XRD patterns of a Zn/CPZ-H//MnO₂ full cell at various voltages. Ex situ high-resolution XPS spectra of **b** Mn 2*p*, **c** Zn 2*p*, **d** O 1*s*, and **e** N 1*s* at various charge–discharge statuses. **f**, **g** In situ Raman spectrum of a Zn/CPZ-H//MnO₂ full cell. **h** Schematic illustration of reversible Zn²⁺ intercalation/de-intercalation in a MnO₂ cathode during an electrochemical process.

de-intercalation during the electrochemical process is illustrated in Fig. 5h. The promotion of the two-electron reduction (Mn⁴⁺ to Mn²⁺) caused the MnO₂ cathode to increase its storage of Zn²⁺ during the electrochemical process, which provided additional energy for storage in the Zn/MnO₂ system.

### Sustainability of the CPZ-H electrolyte in a Zn/MnO₂ full cell

The sustainable biomass composition of the CPZ-H electrolyte was explored by examining the recyclability and degradability of the CPZ-H electrolyte in a Zn/MnO₂ battery. As shown in Fig. 6a, the aged CPZ-H

electrolyte was dried and grinded into powder, and then was treated with a few drops of water to form a solution and transferred into molds, which were heated to obtain recovered CPZ-H (rCPZ-H). Owing to the high powder healing ability of this biomass hydrogel, it can be recovered by a simple process, which provides a good feasibility verification for its large-scale recycling technology. To survey the electrochemical performance of rCPZ-H electrolytes, we combined Zn foil with a rCPZ-H (Zn/rCPZ-H) symmetric cell. The resulting Zn/rCPZ-H// Zn/rCPZ-H symmetric cell operated continuously over 750 h at a current density of 5 mA cm⁻² and a limited capacity (10 mAh cm⁻²; Fig. 6b),

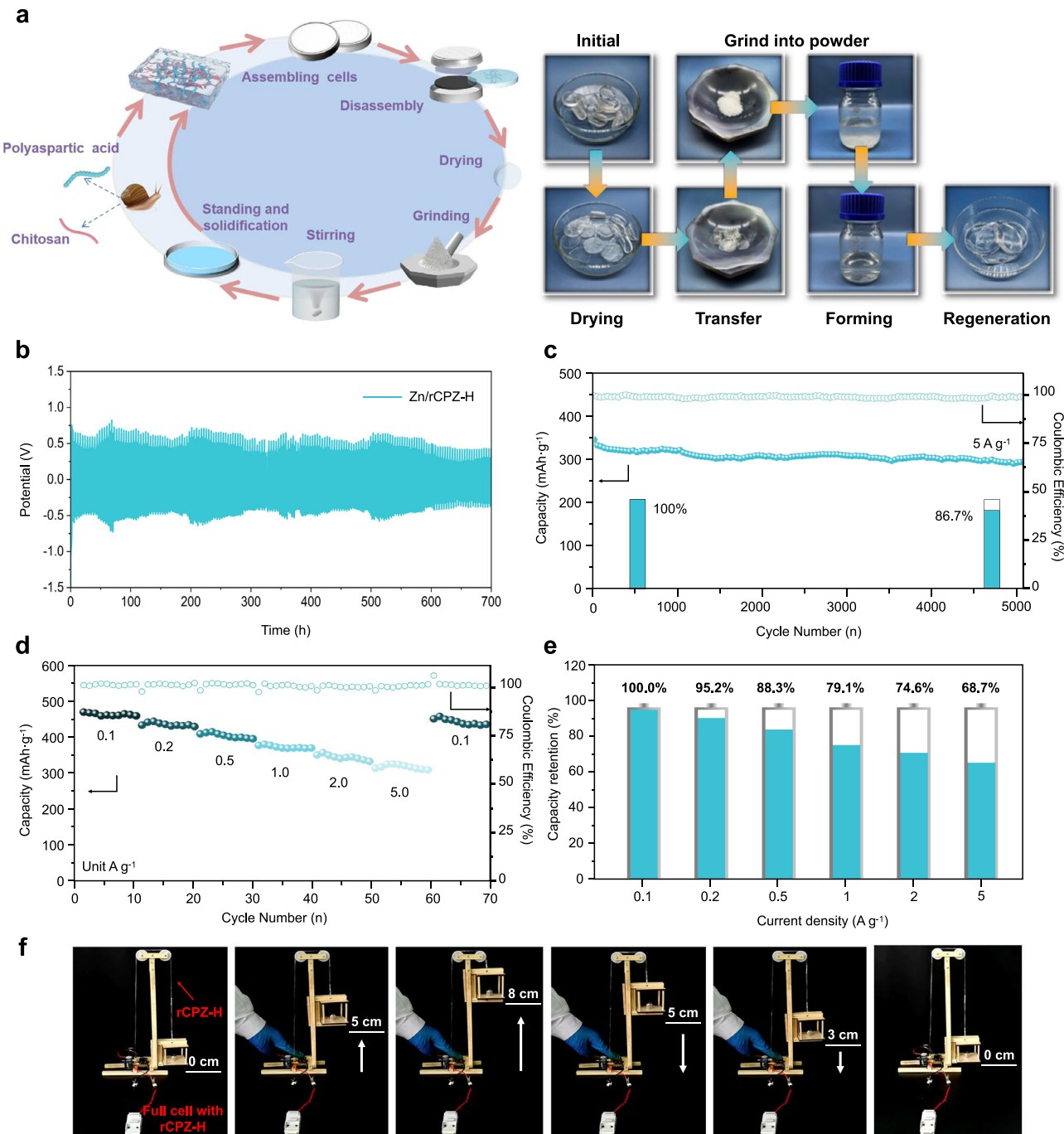

**Fig. 6 | Recyclable properties of the CPZ-H electrolyte in a Zn/MnO₂ full cell.**
**a** Schematic diagram of a CPZ-H electrolyte recycling process and optical photographs of various stages in the actual recycling process. **b** Voltage profiles of symmetric cells with Zn/rCPZ-H anodes. **c** Cyclic performance of a Zn/CPZ-H// MnO₂ full cell at a current density of 5 A g⁻¹. **d**, **e** Rate performance and capacity retention of a Zn/rCPZ-H//MnO₂ full cell. **f** Photographs of an elevator device powered by a Zn/rCPZ-H//MnO₂ full cell.

which revealed that the rCPZ-H electrolyte retained high utilization value. Therefore, the rCPZ-H electrolyte was also used in an aqueous-acid Zn/MnO₂ full battery, which was found to exhibit an excellent cycling performance: a capacity retention of 86.7 % after 2,400 cycles at a current density of 5 A g⁻¹ (Fig. 6c). Furthermore, the rate performance of the Zn/rCPZ-H//MnO₂ full cell at various current densities (Fig. 6d, e) shows that it had capacity retention of 68.7% at a high current density of 5 A g⁻¹. Moreover, the Zn/rCPZ-H//MnO₂ full cell achieved an average capacity of 453.6 mAh g⁻¹ when the current density was switched back to 0.1 A g⁻¹ after 60 cycles, and a high CE (>97.6%), demonstrating the good reversibility of the rCPZ-H

electrolyte. To demonstrate the practical application of the Zn/rCPZ-H//MnO₂ full cell, two cells were connected in series (Supplementary Fig. 22) to power a small elevator, which enabled it to rapidly lift a weight a height of 8 cm (Fig. 6f and Supplementary Movie 1). To evaluate the mechanical performance of the rCPZ-H electrolyte in practical applications, it was molded into a traction rope, which was then used in the small elevator. The structure of this rope was largely undamaged during the operation of the elevator, even during repeated operations. In addition, the rCPZ-H electrolyte could be used in the form of a conveyor belt on a motor to maintain the high-speed operation of a small Ferris wheel (Supplementary Fig. 23 and

Supplementary Movie 2), which showcases the excellent scratch resistance and toughness of the rCPZ-H electrolyte, owing to its dynamic and reversible hydrogen-bonding and ionic complexation capacity.

In addition, the CPZ-H electrolyte is biodegradable, as its key structural components are CS and PASP, so it can be directly biodegraded after recycling and reuse. Specifically, as diagrammed in Fig. 7a, the CPZ-H electrolyte could be degraded into plant nutrients by bacteria and microorganisms after being buried in the soil. To verify the feasibility of this process, a small piece of CPZ-H electrolyte was placed in a *Bacillus* culture fluid, which was maintained at a constant temperature and humidity. The CPZ-H electrolyte was gradually decomposed and absorbed by the *Bacillus*, and almost disappeared over a 90-day incubation (Fig. 7b–e). Moreover, the distribution of bacteria on the CPZ-H surface at different periods was observed by microscope. With increasing incubation time, the CPZ-H electrolyte was effectively decomposed and absorbed by the *Bacillus*, and the nutrients they obtained promoted their continued growth on the surface of the CPZ-H electrolyte (Fig. 7f–i). Thus, ultimately, the constituents of the CPZ-H electrolytes would be naturally released into the environment (Fig. 7j) by microorganisms and would serve as fertilizer for plants. Overall, this green microbial degradation method exhibited a satisfactory degradation rate as compared to those that have been previously reported for biodegradable energy-storage devices (Fig. 7k)[22,38–42].

## Discussion

In summary, an innovative biomass-based electrolyte, the CPZ-H electrolyte, was fabricated from CS and PASP and found to effectively inhibit side reactions and the uncontrolled formation of Zn dendrites. Given the numerous anionic functional groups of CS and PASP, with their high adsorption energy for the Zn (002) crystal plane, it was proposed that a synergistic mechanism within the CPZ-H electrolyte accounted for this inhibition. This mechanism involves equilibrium ion flux and preferred horizontal orientation of growth of the crystal plane on the Zn anode (an E-P mechanism). Therefore, the Zn/CPZ-H//Zn/CPZ-H symmetric cell had a high cyclic performance (2200 h) with excellent reversibility at 10 mA cm$^{-2}$. Due to the excellent H$^+$-redistribution and desolvation ability of the CPZ-H electrolyte in an acidic environment, an aqueous-acid Zn/CPZ-H/MnO$_2$ full cell based on a two-electron redox reaction achieved a large capacity (523.6 mAh g$^{-1}$) and excellent cyclic performance, with a capacity retention of 92.5% after 5000 cycles. Moreover, the CPZ-H electrolyte is sustainable and has high recycling usability, creating infinite possibilities for the development of green batteries. This study has established a significant path toward the development of multifunctional and sustainable biomass electrolytes for use in high-performance green and aqueous energy-storage devices.

## Methods

### Preparation of CPZ-H electrolytes and Zn/CPZ-H anode

About 7 g NaOH and 14 g urea were dissolved into deionized water of 80 mL to obtain the alkaline solvent. Then chitosan of 5 g, polyaspartic acid of 0.8 g were dispersed in the above alkaline solvent; after stirring thoroughly for 30 min at room temperature and, 2 M ZnSO$_4$ solution of 10 mL was slowly dropped the above solution. The mixture solutions was placed at 4 °C for 3 h and then at −20 °C overnight in a cold environment. Afterward, a frozen solution was taken out and stirred lightly until clear solutions were obtained. The mixture solutions of CS/PASP/ZnSO$_4$ were injected into a glass mold (the size of the mold was 15 mm × 15 mm × 0.2 mm) and gelled by heating at 60 °C for 1.5 h. Finally, the CPZ-H electrolyte were obtained by soaking the gels in deionized water until the value of pH closes to 7.0. For the CP-H electrolyte, the preparation process is similar to the CPZ-H electrolyte except that without the addition of ZnSO$_4$. For the preparation of Zn/CPZ-H anode, the above mixture solution were injected into a mold

and CPZ-H electrolyte with a thickness of ~0.15 mm was achieved by heating at 60 °C for 1.5 h and soaking the gels in deionized water until the value of pH closes to neutral. Afterward, CPZ-H with high adhesion is attached to the surface of Zn foil, and the CPZ-H/Zn anode with a diameter of 16 mm was obtained by cutting Zn foil covered by CPZ-H.

### Synthesis of MnO$_2$ cathode materials

KMnO$_4$ of 1.2 g and MnSO$_4$ of 0.5 g were added into 60 mL deionized water under stirring for 30 min at room temperature until the mixture solutions become dark red solutions. And then above mixture solutions were transferred into a 100 mL Teflon-lined autoclave. The autoclave was then maintained at 160 °C for 4 h to obtain MnO$_2$ powder, and the product was washed with deionized water and ethanol three times and vacuum-dried for 12 h.

### The assembling of aqueous Zn/MnO$_2$ coin cells

As-prepared MnO$_2$ powder, super P powder, and polyvinylidene fluoride (PVDF) with a weight ratio of 7:2:1 were successively dispersed in 1-methyl−2-pyrrolidinone (NMP) solvent meanwhile continually stirred for 12 h. Subsequently, the formed slurry was evenly coated on stainless steel foil (thickness was ≈0.01 mm) and vacuum-dried at 60 °C overnight, and cut into disks with a diameter of 16 mm as a cathode. The MnO$_2$ cathode was obtained with a mass loading of 1.82 mg cm$^{-2}$. Next, the pretreated CPZ-H electrolyte is realized by the following method, The mixture solutions of CS/PASP/ZnSO$_4$ were injected into a glass mold (the size of the mold was 15 mm × 15 mm × 0.2 mm) and dried at 60 °C for 1.5 h to allow complete cross-linking polymerization. Subsequently, the as-prepared CPZ-H electrolyte were soaked in deionized water until the value of pH closes to 7.0. And then, the CPZ-H electrolyte was immersed in H$_2$SO$_4$ solution (PH = 1) and H$_2$SO$_4$ solution (PH = 2), respectively. The CPZ-H, after fully absorbing protons, was attached to the Zn foil and cut into a diameter of 16 mm round as an anode. Finally, MnO$_2$ cathode, Zn foil anode, glass fiber as the separator (Whatman, D1823-025), and protonated CPZ-H electrolyte were assembled in Coin-type cells (CR2032).

### Regeneration of CPZ-H electrolyte

Firstly, the waste cell was disassembled, and the Zn/CPZ-H anode was soaked in water until the gel electrolyte was completely peeled off. Afterward, the regenerated CPZ-H electrolyte was dried at 95 °C to completely dehydrate and the as-obtained dried gel was grinded into hundred-micrometer fine powder. Then, the dried gel powder was placed in water and continuous stirring for several hours to promote fine powder dissolution. Subsequently, the dissolved electrolyte powder is transferred to the mold, the regenerated CPZ-H electrolyte can be obtained after further heating process at 70 °C.

### Characterizations and electrochemical measurements

XRD was performed by Bruker-AXS micro diffractometer (D8 Advance) with Cu K radiation to determine the phase of samples. The morphologies and microstructures of samples were characterized using the field emission scanning electron (FESEM, Sigma 500 Zeiss) as well as transmission electron microscopy (TEM, Tecnai G2 F20 S-TWIN). The characterization analysis of hydrogel electrolytes was conducted using Fourier-Transform infrared spectroscopy (NICOLET iS50) and nitrogen sorption isotherms (ASAP 2020 volumetric adsorption analyzer). The morphology of the Zn anode was surveyed by a 3D Laser Confocal Scanning Microscope (LCSM, KEYENCE VK-X200). The X-ray photoelectron spectroscopy (XPS) of electrode materials were implemented by an ESCALab MKII X-ray photoelectron spectrometer. All of the electrochemical tests were carried out at room temperature. Electrochemical characterization of symmetrical cells with bare Zn, Zn/CZ-H, and Zn/CPZ-H anode using conducted using both transparent cells and 2032-type coin cells. Moreover, Zn/Zn symmetric cells and Zn/Cu asymmetric cells are used for

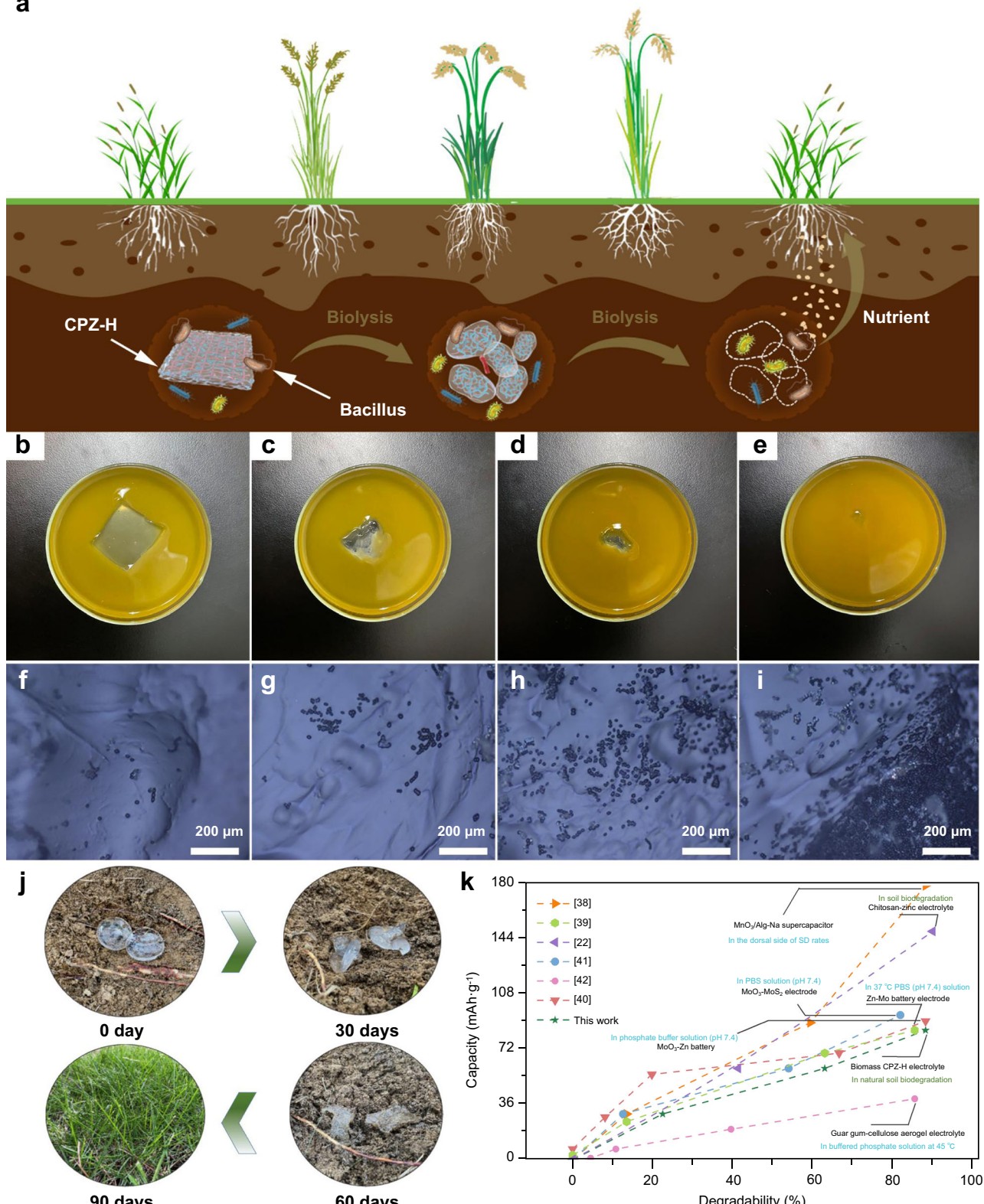

**Fig. 7 | Degradability of the CPZ-H electrolyte in a Zn/MnO₂ full cell.**
**a** Diagrammatic sketch of the CPZ-H electrolyte biodegradation process.
**b**–**e** Optical photographs of the decomposition process of a CPZ-H electrolyte in a *Bacillus* culture liquid. **f**–**i** Microscopic photographs of *Bacillus* on the CPZ-H electrolyte at different stages of decomposition. **j** Photographs of actual decomposition of the CPZ-H electrolyte in natural soil at various timepoints.
**k** Comparison of CPZ-H electrolyte biodegradation mechanism, time, and characteristics with those that have been reported for other biodegradable energy-storage devices[22,38–42].

plating-stripping tests. Plating-stripping profiles and galvanostatic charge–discharge curves were taken from the Neware CT4000 battery testing system. Linear polarization and cyclic voltammetry (CV) tests were carried out on an electrochemical (CHI660E, Shanghai, China) at the voltage range of −0.7 to −1.2 V vs. Zn/Zn$^{2+}$. For the CPZ-H/Zn//MnO$_2$ full cell, the galvanostatic charged/discharged test of the full cell was carried out in a voltage range of 0.8−2.0 V at different current densities. The electrochemical impedance spectroscopy (EIS) was collected by an electrochemical workstation at a frequency range of 0.01 Hz to 100 kHz. The Zn$^{2+}$ diffusion coefficients of the full cells were executed by galvanostatic intermittent titration technique (GITT) at a current density of 5 A g$^{-1}$.

## Computational methods

Vienna Ab initio Simulation Package (VASP)[43] with the projector augmented wave (PAW) method is used for the surface of Zn (002) surface and the hydrogen adsorption free energy $\Delta G_{H^*}$. The interaction of Zn ions with water molecules and other molecules is processed using the B3LYP/6-31 G* method by Gaussion09 software package[44]. The optimized structures of bulk Zn ($a = b = 2.585$ Å, $c = 4.983$ Å, $\alpha = \beta = 90°$, $\gamma = 120°$) was selected, and then a $5 \times 5$ lateral periodicity of Zn (002) surface was constructed to study the interaction with molecules. The bottom two layers of atoms in the Zn slab are fixed in all surface models. In addition, two times, chitosan monomers and minimal polyaspartic acid monomers were used to simulate their macromolecular structures, respectively. The exchange-functional is treated with the Perdew-Burke-Ernzerhof (PBE)[45] functional, combined with the DFT-D3 correction[46]. The cut-off energy of the plane wave base was installed at 400 eV and a Monkhorst-Pack grid of $2 \times 2 \times 1$ for the Zn surface models was used. The convergence energy threshold of $10^{-5}$ eV was applied in the self-consistent calculations. The optimization of equilibrium geometries and lattice constants was in pursuance with the condition that the maximum stress on each atom was within 0.03 eV/Å.

## Data availability

All data that support the findings of this study are provided within the paper and its Supplementary Information. All additional information is available from the corresponding authors upon request. Source data are provided with this paper.

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

## Acknowledgements

G.Z. appreciates the support from the National Key Research and Development Program of China (2019YFA0705700), National Natural Science Foundation of China (U21A20174), Guangdong Innovative and Entrepreneurial Research Team Program (2021ZT09L197), Shenzhen Science and Technology Program (KQTD20210811090112002), the Start-up Funds, Interdisciplinary Research and Innovation Fund of Tsinghua Shenzhen International Graduate School, and the Tsinghua Shenzhen International Graduate School-Shenzhen Pengrui Young Faculty Program of Shenzhen Pengrui Foundation (SZPR2023007). J.Z. acknowledges financial support from the Research Grants Council of Hong Kong (RGC Postdoctoral Fellowship Scheme, Grant No.: PDFS2122-5S03).

## Author contributions

G.Z., B.X., and J. Z. supervised the research. H.L. and J.Z. conceived the research. H.L., J.H., X.W., and K.Z. carried out the experiments, and collected and analyzed the experimental data. X.X., Q.H., and J.Y. gave helpful advice on manuscript preparation. H.L. and K.Z. wrote the paper. H.L., J.H., X.W., and K.Z. were equal major contributors. All authors participated in discussions of the research.

## Competing interests

The authors declare no competing interests.
