## [Peer Review File · Nature Communications]

A Recyclable Biomass Electrolyte Towards Green Zn-Ion BatteriesREVIEWER COMMENTS

Reviewer #1 (Remarks to the Author):

In this work authors show a biobased electrolyte for zinc ion batteries. The approach for electrolyte synthesis is novel and interesting, and the battery performance is carefully studied. Many characterization techniques are applied, and the fundamentals are well studied. Obtained results outperform most of the reports in literature, and the work is completed with degradation studies. This manuscript may be of interest for the readers of the journal. However, there are certain issues that should be addressed before the manuscript can be accepted for publication.

- Authors use “functional hydrogel electrolyte”; however, its definition remains ambiguous. Why functional is added here? Please, define.
- The Introduction section must state clearly the motivation behind the selection of chitosan and polyaspartic acid among all the biopolymers available. Is the combination of the functional groups of these materials especially interesting for ion conduction? What about water solubility? And mechanical properties?
- Authors use the term SUSTAINABLE in the title. However, they do not provide environmental impact metrics. Therefore, I suggest avoiding this term, at least in the title.
- Caption for Figure S1: it should make clear that this is for the freeze-dried sample.
- Which is the motivation behind MnO₂ cathode selection? It is more recyclable? MnO₂ can suffer a marked capacity decay after cycling, so if aiming at batteries with low environmental impact, other materials may be preferred,
- The CPZ-H electrolyte can be degraded into plant nutrients by bacteria and microorganism after burying in soil as shown in Figure 7. However, what about the whole battery? Other reports have already shown degradable zinc-based energy storage devices, so, which is the actual contribution here? Authors must include a section to compare the degradation mechanics, time and characteristics here observed with literature (regarding disposable, compostable or degradable zinc ion batteries).
- Similarly, how the electrolyte has been recycled remains poorly explained in the manuscript. Please, provide further details on the procedure, and summarize the state-of-the-art in this context.
- Graphics are very visual; however, including many panels into a single Figure can make the understanding difficult. Authors are encouraged to revise this aspect.
- Has the CS and PASP ratio in the hydrogel been optimized? What about changing to low and high CS concentrations, for example?

Reviewer #2 (Remarks to the Author):

This is an interesting work however there are few questions or comments that I would like authors to address

1. To prepare electrolyte, you mentioned you precooled CS-PSAP solution followed by heating and stirring: what was the assumption or science behind it?
2. If you are using gel electrolyte then why you are using coin cell and not pouch cell, please explain
3. On Page 5, before performing SEM why did you freeze cast the electrolyte?
4. On Page 5, line 106, you mentioned extensive layered and interconnected pores and high surface area of the electrolyte. How did you obtain that for CPZ-H?
5. On page5, you mentioned CPZ-H have high water retention and more hydrophilicity. What is the benefit of having these two properties.
6. What is the thickness of your gel electrolyte layer?
7. What is the importance of having high ion diffusion coefficient of electrolyte and how water affects ion diffusion coefficient?

8. Did you measure the ionic conductivity of CPZ-H and CZ-H? if not please do so and discuss how it will contribute to the kinetics of the cell.
9. In Fig2(a) what is the significance of having voltage hysteresis? Is it good to have low voltage hysteresis if yes please explain why?
10. In Fig 2(a), how are you measuring the voltage hysteresis
11. What is desolvation energy barrier? What factors are contributing to desolvation energy barrier ? Why it is significant for Zn anode?
12. What is the average operating voltage of proposed cell?
13. The specific capacity you have mentioned for Zn/MnO₂ is coming from two electron reduction. However, your CV graph in Fig.4 shows/suggest only one reduction peak. Potential vs specific capacity also shows one electron reduction only. Please prove how you are getting second electron reduction in your system

Detailed Responses to Referees' Comments

We highly appreciate the editor and reviewers for your valuable time and effort in handling, reviewing and commenting on our manuscript. We think these suggestions are very helpful for our present and future work. We have carefully considered all the comments and revised our manuscript accordingly. All the revisions are highlighted in the revised manuscript. The detailed comments and responses are listed as below:

Reviewer #1:

In this work authors show a biobased electrolyte for zinc ion batteries. The approach for electrolyte synthesis is novel and interesting, and the battery performance is carefully studied. Many characterization techniques are applied, and the fundamentals are well studied. Obtained results outperform most of the reports in literature, and the work is completed with degradation studies. This manuscript may be of interest for the readers of the journal. However, there are certain issues that should be addressed before the manuscript can be accepted for publication.

Response: We are grateful for the reviewer's kind comments and suggestions. We have carefully revised the manuscript according to your kind comments and suggestions.

Summary of the comments

1) *Authors use "functional hydrogel electrolyte" ; however, its definition remains ambiguous. Why functional is added here? Please, define.*

Response: We are grateful for the reviewer's comments and suggestions. In fact, functional hydrogel electrolyte means that the as-prepared hydrogel electrolyte possesses different functional characteristics in this work, including satisfactory mechanical properties, high water retention ability, good resistance to acid corrosion and inhibition of Zn dendrite. Benefitting from multi-functional properties of this hydrogel electrolyte, the Zn/CPZ-H//Zn/CPZ-H symmetrical cell can effectively promote Zn anode to achieve ultralong cycle life of 2,200 h at high current density. Meanwhile, the CPZ-H electrolyte can trigger two-electron redox of the Zn/MnO₂ system to achieve a higher discharge capacity.

2) *The Introduction section must state clearly the motivation behind the selection of chitosan and polyaspartic acid among all the biopolymers available. Is the combination of the functional groups of these materials especially interesting for ion conduction? What about water solubility?*

And mechanical properties?

Response: Thank you very much for your suggestions. The motivation behind the selection of chitosan (CS) and polyaspartic acid (PSAP) include two aspects: (1) Compared with other hydrogel ingredients, CS and PSAP as naturally biomass materials have been widely found in the shells of mollusks such as snails, which possess good recyclability and biodegradability; and (2) CS and PASP skeleton with abundant carboxy and amino that can be well crosslinked with hydrogen bond to effectively enhance the mechanical properties of hydrogel. Please kindly refer to Page 3, line 71 to line 75 in the revised manuscript. The coupling of carboxyl and amino groups occurred in the CPZ-H electrolyte through hydrogen bonds. This chemical crosslinking accelerates the coupling of CS and PASP dual networks in hydrogel, which can reduce water solubility of CPZ-H electrolyte and improve mechanical properties. Meanwhile, carboxyl as an ionizable group can further improve the ionic conductivity of the hydrogel (*Trends in Chemistry*, 2019, 1, 335-348).

3) Authors use the term SUSTAINABLE in the title. However, they do not provide environmental impact metrics. Therefore, I suggest avoiding this term, at least in the title.

Response: Thank you very much for your comments. We have corrected title in the revised manuscript as suggested.

4) Caption for Figure S1: it should make clear that this is for the freeze-dried sample.

Response: Thank you so much for your comments. We have corrected this mistake.

5) Which is the motivation behind MnO₂ cathode selection? It is more recyclable? MnO₂ can suffer a marked capacity decay after cycling, so if aiming at batteries with low environmental impact, other materials may be preferred.

Response: Thank you very much for your valuable comments. MnO₂ as cathode material of zinc ion battery has many advantages such as simple preparation, low cost and high safety, thus Zn-MnO₂ battery system is a kind of large-scale rechargeable battery with broad application prospects. However, in practical application, the valence state of Mn in traditional natural water electrolyte only changes from Mn⁴⁺ to Mn³⁺ (theoretical capacity = 308 mAh g⁻¹), which leads

to that the actual discharge capacity of MnO₂ cathode material is not high. Encouragingly, MnO₂ experiences a two-electron-transfer redox reaction in acidic Zn-MnO₂ system, which can promisingly double the specific capacity of MnO₂ from 308 to 616 mAh g⁻¹ (*Nat. Energy*, 2018, 3, 428). However, acidic electrolyte with H⁺ will cause strong hydrogen evolution reaction on the surface of Zn anode, which greatly affects the cyclic life of the acidic Zn-MnO₂ battery. To overcome this problem, in this work, we prepared CPZ-H electrolyte, which has good resistance to acid corrosion as well as high inhibition on Zn dendrite and hydrogen evolution reaction. Therefore, CPZ-H electrolyte in acidic Zn-MnO₂ battery not only triggers two-electron-transfer redox reaction to obtain high discharge capacity, but also suppresses side reaction of Zn anode to achieve long cyclic life of battery. Please refer to Page 4 line 90-93.

6) The CPZ-H electrolyte can be degraded into plant nutrients by bacteria and microorganism after burying in soil as shown in Figure 7. However, what about the whole battery? Other reports have already shown degradable zinc-based energy storage devices, so, which is the actual contribution here? Authors must include a section to compare the degradation mechanics, time and characteristics here observed with literature (regarding disposable, compostable or degradable zinc ion batteries)

Response: Thank you very much for your comments. This work focuses on a new biomass electrolyte originates from natural biomass materials, which can achieve simultaneously suppress zinc dendrites and trigger two-electron-transfer redox reaction of MnO₂ in Zn-MnO₂ battery. Therefore, we mainly investigated the recoverability and biodegradability of this CPZ-H electrolyte. Actually, the CPZ-H electrolyte still has a high zinc dendrite inhibition effect after recovery (Figure 6b), and it can also be turned into pollution-free fertilizer under the degradation of bacillus. Indeed, other reports have reported degradable zinc-based energy storage devices, but most of them are carried out under specific stability and environment. For example, Karami-Mosammam et al. achieved the degradation of MoO₃-Zn battery in phosphate buffer solution (pH = 7.4) (*Adv. Mater.* 2022, 34, 2204457). Sheng et al. reported the degradation process of MoO₃-Alg-Na supercapacitor in PBS solution at 85 °C (*Sci. Adv.* 2021, 7, eabe3097). The degradation mechanism of materials is the process in which the internal polymer segments is broken into low molecular weight oligomers under specific conditions,

and finally decomposed into monomers. Different from above degradation mechanism, the biomass CPZ-H electrolyte can be degraded by microorganisms. The main mechanism in this work is that the organic material in the waste CPZ-H electrolyte is transformed into microbial cell material to provide energy for its own life activities and synthesis of new cell material, and the other part is decomposed into simple inorganic material (such as carbon in the organic matter is oxidized into carbon dioxide, hydrogen is oxidized into water, and nitrogen is oxidized into ammonia, nitrite and nitrate) to provide nutrients for plants in the soil. For the pure biodegradability of full Zn-based battery, we will further conduct the investigation in the future work. In the revised manuscript, we have supplemented the comparison of degradation mechanism, time and characteristics with the previously reported degradable energy storage devices as shown in Figure 7k. Overall, this green microbial degradation method exhibited a satisfactory degradation rate as compared to those that have been previously reported for biodegradable energy-storage devices. Please kindly refer to Page 15, line 423 to 425 in revised manuscript.

Figure 7k. Comparison of degradation mechanism, time and characteristics with the previously reported degradable energy storage devices.

7) Similarly, how the electrolyte has been recycled remains poorly explained in the manuscript. Please, provide further details on the procedure, and summarize the state-of-the-art in this context.

Response: Thank you very much for your valuable comments. We provided the information on electrolyte recycled process in Supporting information (Experimental Section 1.4). Firstly, the waste cell was disassembled, meanwhile the Zn/CPZ-H anode was soaked in water until gel electrolyte was completely peeled off. Afterwards, regenerated CPZ-H electrolyte was dried at 95 °C until to completely dehydrate and the as-obtained dried gel was grinded into hundred-micrometer fine powder. Then, the dried gel powder was placed in water and continuous stirring for several hours to promote fine powder dissolution. Subsequently, the dissolved electrolyte powder is transferred to the mold, and the regenerated CPZ-H electrolyte can be obtained after further heating process at 70 °C. Owing to the high powder healing ability of this biomass hydrogel, it can be recovered by simple process, which provides a good feasibility verification for its large-scale recycling technology. We have supplemented the details of regeneration of CPZ-H electrolyte: Firstly, the waste cell was disassembled, and the Zn/CPZ-H anode was soaked in water until gel electrolyte was completely peeled off. Afterwards, regenerated CPZ-H electrolyte was dried at 95 °C to completely dehydrate and the as-obtained dried gel was grinded into hundred-micrometer fine powder. Then, the dried gel powder was placed in water and continuous stirred for several hours to promote fine powder dissolution. Subsequently, the dissolved electrolyte powder is transferred to the mold, the regenerated CPZ-H electrolyte can be obtained after further heating process at 70 °C. Please refer to Page 3 line 26-line 31 in revised supporting information. Meanwhile, the advanced nature of this technology is summarized in revised manuscript as follows: As shown in Figure 6a, aged CPZ-H electrolyte was dried and grinded into powder, and then was treated with a few drops of water to form a solution and transferred into moulds, which were heated to obtain recovered CPZ-H (rCPZ-H). Owing to the high powder healing ability of this biomass hydrogel, it can be recovered by simple process, which provides a good feasibility verification for its large-scale recycling technology. Please refer to Page 14, line 386 to line 391 in the revised manuscript.

8) Graphics are very visual; however, including many panels into a single Figure can make the understanding difficult. Authors are encouraged to revise this aspect.

Response: Thank you very much for your advice. In fact, each chapter in this article contains a scientific problem to be studied. And many panels into a single Figure can systematically

explain each scientific problem that need to be solved. In order to make readers understand more intuitively, we have simplified Figure 1-6 and the structure of Figure 7 has been optimized . Please refer to new Figure 1-7 in revised manuscript.

9) *Has the CS and PASP ratio in the hydrogel been optimized? What about changing to low and high CS concentrations, for example?*

Response: Thank you very much for your comments. The CS and PASP ratio in hydrogel has been optimized in this work. For the CPZ-H electrolyte, CS is acted as the main skeleton of CPZ-H electrolyte, which can stabilize its structure. PASP is an additive that can be cross-linked with CS in hydrogel to further improve the mechanical properties of CPZ-H electrolyte. Therefore, the content of CS is much higher than that of PASP, which enable crosslinked CPZ-H electrolyte to have good mechanical properties. We also investigated structural characteristics of hydrogel with different ratios of CS and PASP, which are shown in the following figures and table.

Figure X1. Optical photos of the CPZ-H electrolyte with different concentration ratios of CS and PASP.

TableX1. Shape fidelity of CPZ-H electrolyte with different concentration ratio of CS/PASP

Concentration of CS (%)	80	70	60	50	40
Concentration of PASP (%)	20	30	40	50	60
Shape fidelity of CPZ-H electrolyte	High	High	Medium	Medium	Low

It can be clearly seen that the shape fidelity of CPZ-H electrolyte is greatly reduced when the decrease of CS content in CPZ-H electrolyte from 80 % to 50 %. In addition, the CPZ-H electrolyte could not maintain its shape when the the content of CS was reduced to 40 %. According to the above results, the CS and PASP concentration ratio of 80:20 and 70:30 can

achieve the hydrogel electrolyte with high shape fidelity. Therefore, the suitable hydrogels can be obtained when the concentration of CS is higher than 70 %.

Reviewer #2:

This is an interesting work however there are few questions or comments that I would like authors to address

Response: We appreciate your careful review and constructive suggestions on our manuscript. Listed below is your comments and our replies.

Summary of the comments

1) To prepare electrolyte, you mentioned you precooled CS-PASP solution followed by heating and stirring: what was the assumption or science behind it?

Response: Thank you very much for your valuable comments. The precooled process of CS-PASP solution is used to increase solubility of CS, due to its difficulty to dissolve in aqueous solution. It is found that chitosan can be dissolved in precooled aqueous alkali-urea solutions (just like cellulose), and chitosan can form complex with urea and NaOH at low temperatures, and hence dissolves in the solution (*ChemPhysChem.*, 2007, 8, 1572-1579). The CS solution is completely dissolved after further stirring and becomes a transparent solution. Due to that these solutions are thermosensitive (*Soft Matter*, 2014, 10, 8245-8253), and then the dissolved CS-PASP solution was transferred to the mold for heating at 60 °C. The CS can be crosslinked with PASP to form CPZ-H electrolyte.

2) If you are using gel electrolyte then why you are using coin cell and not pouch cell, please explain.

Response: Thank you very much for your comments. The reason why we use coin cell is that it is easier to disassemble. Actually, the CPZ-H electrolyte is also suitable for pouch cell. We assembled Zn/CPZ-H//Zn/CPZ-H symmetrical pouch cell, which also exhibits stable electrochemical performance over 500 h at current density of 50 mA cm⁻² with a limited capacity of 25 mAh g⁻¹.

Figure X2. (a) Optical photos of the Zn/CPZ-H symmetrical pouch cell, (b) Voltage profiles of Zn/CPZ-H symmetrical pouch cell at current density of 50 mA cm⁻² with limited capacity of 25 mAh g⁻¹.

3) On Page 5, before performing SEM why did you freeze cast the electrolyte?

Response: Thank you very much for your comments. In order to observe the internal structure of hydrogel more clearly, it is necessary to freeze-dry hydrogel in advance to remove its internal moisture and maintain its structural integrity.

4) On Page 5, line 106, you mentioned extensive layered and interconnected pores and high surface area of the electrolyte. How did you obtain that for CPZ-H?

Response: Thank you very much for your comments. The CPZ-H electrolyte was obtained by precooled aqueous alkali-urea solutions (Details in Experimental Procedures 1.1). And then the as-obtained CPZ-H electrolyte were freeze dried in a frozen dryer (LGJ18-A, Shunzhi, China) at temperature of -60 °C and vacuum degree of 15 Pa for at least 48 h. The freeze-drying hydrogel can be obtained after the above steps are completed.

5) On page5, you mentioned CPZ-H have high water retention and more hydrophilicity. What is the benefit of having these two properties.

Response: Thank you very much for your comments. Ion transport of hydrogel mainly depends on its internal solvent. The evaporation of water in hydrogel at room temperature will greatly affect its ionic conductivity in practical battery applications. The CPZ-H with high water retention can ensure its stable ionic conductivity during the operation of the battery. For the hydrophilicity of hydrogel, hydrogel is a kind of polymer with hydrophilic groups, which can be swollen by water but insoluble in water, and it has a three-dimensional network structure.

Polymers with high hydrophilicity can effectively increase the degree of polymerization of hydrogel by chemical or physical crosslinking with each other.

6) *What is the thickness of your gel electrolyte layer?*

Response: Thank you very much for your constructive comments. The mixture solutions of CS/PASP/ZnSO₄ were injected into glass mold (the size of mold was 15 mm × 15 mm × 0.2 mm) and gelled by heating at 60 °C for 1.5 h. The cured CPZ-H electrolyte was obtained, and its thickness was 0.15 mm.

7) *What is the importance of having high ion diffusion coefficient of electrolyte and how water affects ion diffusion coefficient?*

Response: Thank you very much for your constructive comments. The ion diffusion coefficient of electrolyte refers to the diffusion rate of ions in electrolyte, which can directly affect the ion migration between the positive and negative electrodes in the battery operation. The diffusion rate of ions in the electrolyte also determines the charge-discharge rate and energy density of the battery. Hydrogel electrolyte contains a lot of water, and water is also the medium of ion transport in hydrogel electrolyte. Water loss will lead to the extreme decrease of ion diffusion rate in hydrogel electrolyte, and then the battery will fail. Therefore, hydrogel with high water retention can effectively prevent water loss at room temperature.

8) *Did you measure the ionic conductivity of CPZ-H and CZ-H? if not please do so and discuss how it will contribute to the kinetics of the cell.*

Response: Thank you very much for your comments. The ionic conductivity of CPZ-H and CZ-H electrolytes at different temperatures of -20 to 30°C via electrochemical workstation has been exhibited in the new Fig. S2. Details are as follows: EIS spectra of CPZ-H and CZ-H electrolytes at different temperatures of -20 to 30°C were recorded on electrochemical workstation in the frequency range of 10⁵ to 10⁻² Hz to obtain the ohmic resistance. Subsequently, the ionic conductivity can be calculated according to the following equation:

$$\sigma = \frac{L}{AR}$$

Where R indicates ohmic resistance, L represents thickness, and A is area.

Compared with CZ-H electrolyte, the ionic conductivity of CPZ-H is improved by introducing PASP. The reason is that the introduction of PASP into CZ-H hydrogel forms more porous structures due to the cross-linking between them. Meanwhile, the carboxyl groups on the surface of PASP attract counter ions and provide more hopping sites for ion transfer (*Sci. Adv.*, 2019, 5, eaau4238; *Adv. Funct. Mater.*, 2020, 30, 2003430). Thus, introduction of PASP can promote the ionic conductivity in CPZ-H electrolyte. The higher the ionic conductivity of the hydrogel electrolyte demonstrates that the faster the ions move in the hydrogel electrolyte. Therefore, the high ionic conductivity of the hydrogel electrolyte has positive significance for improving the rate performance of the full battery.

We highlight the relationship of ionic conductivity of CPZ-H and CZ-H electrolytes and temperature, and discuss the specific reasons that PASP is introduced into CZ-H electrolyte to boost the kinetics of the cell in the revised version. Details as follows: Figure S2 shows the ionic conductivity of the CPZ-H and CZ-H electrolytes at various temperatures of -20 to 30 °C, and the ionic conductivity of the CPZ-H electrolyte is greater than that of the CZ-H electrolyte due to the presence of PASP in the CPZ-H electrolyte, which allows it to form more porous structures via cross-linking. Moreover, the carboxyl groups on the surface of PASP attract counter-ions, which are served as additional hopping sites for ion transfer. Please refer to Page 6, line 147 to 152 in revised manuscript.

Figure S2. Ionic conductivity of CPZ-H and CZ-H electrolytes at different temperatures.

9) In Fig2(a) what is the significance of having voltage hysteresis? Is it good to have low voltage hysteresis if yes please explain why?

Response: Thank you very much for your comments. Generally, the voltage polarization of symmetrical battery test is also called voltage hysteresis. Voltage polarization of symmetrical battery refers to the potential difference caused by charging and discharging of battery. The positive and negative electrodes in a symmetrical battery are both zinc electrodes, and the movement of zinc ions between the two electrodes needs to overcome the interaction between charges, so voltage polarization will occur. The low voltage hysteresis indicates that its overpotential is small, which is more conducive to the rapid and uniform deposition of zinc (ACS Nano, 2022, 16, 12, 21152).

10) In Fig 2(a), how are you measuring the voltage hysteresis.

Response: Thank you very much for your comments. Voltage hysteresis is obtained via constant current charging and discharging of Zn//Zn symmetrical battery under rated current density and limited capacity. The voltage hysteresis refers to the potential difference generated during the charging and discharging of symmetrical batteries. Therefore, the potential difference between the charge and discharge curves of symmetrical batteries is voltage hysteresis

For example: as shown in Figure X3, the voltage hysteresis of this high-resolution voltage profiles is ~ 96 mV.

Figure X3. High-resolution voltage profiles of Zn/CPZ-H symmetric cells.

11) *What is desolvation energy barrier? What factors are contributing to desolvation energy barrier? Why it is significant for Zn anode?*

Response: Thank you very much for your comments. Zinc ions can easily coordinate with water molecules in aqueous electrolyte to form hydrated zinc ions ($[\text{Zn}(\text{H}_2\text{O})_6]^{2+}$), which greatly affects the deposition kinetics of zinc ions. Therefore, the desolvation energy barrier is the energy of removing water molecules from $[\text{Zn}(\text{H}_2\text{O})_6]^{2+}$. The size of desolvation energy barrier depends on the coordination ability and coordination number of solvent molecules in zinc ion solvent shell. The solvation structure of electrolyte will affect the desolvation energy barrier at the electrode-electrolyte interface, and further influence the zinc deposition behavior. The reduction of the desorption barrier of $[\text{Zn}(\text{H}_2\text{O})_6]^{2+}$ is beneficial to the rapid and uniform deposition of zinc ions on the anode (*Angew. Chem.*, 2022, 134, e202207).

12) *What is the average operating voltage of proposed cell?*

Response: Thank you very much for your comments. The average operating voltage of Zn/CPZ-H/MnO₂ full cell is ~ 1.78 V.

13) *The specific capacity you have mentioned for Zn/MnO₂ is coming from two electron reduction. However, your CV graph in Fig.4 shows/suggest only one reduction peak. Potential vs specific capacity also shows one electron reduction only. Please prove how you are getting second electron reduction in your system.*

Response: Thank you very much for your comments. CV curve shows only a pair of redox peaks that indicates two electron transfers (Mn^{4+} to Mn^{2+}) occur simultaneously at the rated potential, which can well correspond to the GCD curve (Figure 4c). Different from the single electron redox reaction shown in the CV curve of traditional Zn//MnO₂ battery, Zn/CPZ-H/MnO₂ battery can achieve redox reaction at a high voltage of 1.8 V, and provide a wider discharge platform and higher discharge capacity. Besides, it is proved from two aspects that the battery has a two-electron redox reaction: (1) An ideal chemical reaction process of acidic aqueous Zn/MnO₂ battery with two-electron redox delivers a high theoretical capacity of 616 mA h g⁻¹ and achieves the maximization of energy density of Zn/MnO₂ battery. And Zn/CPZ-

H/MnO₂ full cell delivers a high capacity of 523.6 mAh g⁻¹ at first cycle, and the discharge capacity of second cycle also reaches 516.4 mAh g⁻¹, which is higher than that of Mn⁴⁺ to Mn³⁺ in the conventional natural aqueous electrolyte (theoretical capacity 308 mAh g⁻¹, *Adv. Energy Mater.*, 2022, 12, 2103705); and (2) we also carried out a systematic ex-situ XPS test on the charging and discharging process of the battery. (Figure 5b). Figure 5b presents the spectrum of Mn 2p during charge/discharge process, and the change of Mn valence can be further illustrated by the spectra of Mn 2p at the different voltage states. The peak intensities of Mn⁴⁺ peaks at 654.7 eV and 642.8 eV are gradually weakened while Mn²⁺ peaks at 653.4 eV and 642.5 eV are enhanced after full cell was firstly discharged to 0.8 V, which further demonstrates a notable valence changing of manganese from Mn (IV) to Mn (II). Afterwards, Mn⁴⁺ peaks regained their initial intensity and Mn²⁺ peak gradually weakened when full cell was charged to 2.0 V. Finally, the intensity of Mn²⁺ can be restored when the full cell is discharged to 0.8 V again, revealing that the two-electron redox occurred during the discharge/charge process (*Adv. Energy Mater.*, 2021, 11, 2102055).

REVIEWERS' COMMENTS

Reviewer #1 (Remarks to the Author):

The biobased electrolyte here constructed not only provides a sustainable path towards energy storage systems but also solves some of the most limiting issues found in zinc-ion batteries. The work is coupled with extensive experimental work to understand the mechanisms by which the synthesized hydrogel electrolyte suppresses the undesired Zn growth. Besides, the end-of-life of the electrolyte is optimized, providing new avenues towards environmentally sustainable batteries.

Figure quality is very good, and the document is well written. Suggested revisions have been well-addressed by the authors and the manuscript now provides solid evidences on the potential of developed materials to inhibit the side-reactions occurring in zinc ion batteries. Accordingly, I suggest accepting this manuscript for publication with no further change.

In addition, I am happy to revise the comments provided to referee #2.

After carefully reading the manuscript, SI and the response letter, I think the authors response is accurate and solid. Authors have performed additional tests and new relevant information is shown in the SI. Moreover, provided responses are well constructed with adequate references. Accordingly, I suggest accepting this work for publication with no further modification.

Detailed Responses to Referees' Comments

Reviewer #1:

The biobased electrolyte here constructed not only provides a sustainable path towards energy storage systems but also solves some of the most limiting issues found in zinc ion batteries. The work is coupled with extensive experimental work to understand the mechanisms by which the synthesized hydrogel electrolyte suppresses the undesired Zn growth. Besides, the end-of-life of the electrolyte is optimized, providing new avenues towards environmentally sustainable batteries. Figure quality is very good, and the document is well written. Suggested revisions have been well-addressed by the authors and the manuscript now provides solid evidences on the potential of developed materials to inhibit the side-reactions occurring in zinc ion batteries. Accordingly, I suggest accepting this manuscript for publication with no further change.

Response: Thank you very much for your careful review and acceptance of our article.

Reviewer #2:

After carefully reading the manuscript, SI and the response letter, I think the authors response is accurate and solid. Authors have performed additional tests and new relevant information is shown in the SI. Moreover, provided responses are well constructed with adequate references. Accordingly, I suggest accepting this work for publication with no further modification.

Response: We thanks for your review and acceptance of our article.